# Continuous Exposure Learning for Low-light Image Enhancement using Neural ODEs

**Donggoo Jung**[*1]**, Daehyun Kim**[*1]**, Tae Hyun Kim**[2†]
Dept. of Artificial Intelligence[1], Dept. of Computer Science[2], Hanyang University
`{dgjung, daehyun, taehyunkim}@hanyang.ac.kr`

## Abstract

Low-light image enhancement poses a significant challenge due to the limited information captured by image sensors in low-light environments. Despite recent improvements in deep learning models, the lack of paired training datasets remains a significant obstacle. Therefore, unsupervised methods have emerged as a promising solution. In this work, we focus on the strength of curve-adjustment-based approaches to tackle unsupervised methods. The majority of existing unsupervised curve-adjustment approaches iteratively estimate higher order curve parameters to enhance the exposure of images while efficiently preserving the details of the images. However, the convergence of the enhancement procedure cannot be guaranteed, leading to sensitivity to the number of iterations and limited performance. To address this problem, we consider the iterative curve-adjustment update process as a dynamic system and formulate it as a Neural Ordinary Differential Equations (NODE) for the first time, and this allows us to learn a continuous dynamics of the latent image. The strategy of utilizing NODE to leverage continuous dynamics in iterative methods enhances unsupervised learning and aids in achieving better convergence compared to discrete-space approaches. Consequently, we achieve state-of-the-art performance in unsupervised low-light image enhancement across various benchmark datasets. Code is available at https://github.com/dgjung0220/CLODE.

## 1 Introduction

Images taken in various low-light environments suffer from insufficient light, leading to the capture of limited information by the camera's image sensor. Therefore, many studies have been conducted to improve the quality of the low-light images and achieve images with optimal exposure levels. In particular, recent supervision-based deep learning approaches (Wang et al., 2022b; Cai et al., 2023; Hou et al., 2023) have shown remarkable performance in enhancing low-light images. However, the process of collecting pairs of low-light scenes and their corresponding ground-truth images for supervised learning is time consuming and resource intensive. As a result, unsupervised approaches that rely solely on low-light images have been proposed to address this problem.

Among many unsupervised low-light image enhancement approaches, curve-adjustment-based methods, conventionally used in photo editing software (*e.g.,* Photoshop), have received much attention.

After the introduction of first learning-based curve-adjustment work by Yuan and Sun (Yuan & Sun, 2012), iterative curve-adjustment-based methods have been explored in various subsequent studies. These unsupervised methods achieve enhancement without using the ground-truth images by fitting the brightness values of pixels in the input image to specific curves. In addition, it is advantageous to preserve local structural information adaptively by allowing efficient pixel-by-pixel computations. For example, ZeroDCE (Guo et al., 2020; Li et al., 2021) introduced a fast and lightweight neural network to predict pixel-wise curve parameter maps within a fixed iteration step. In addition, ReLLIE (Zhang et al., 2021b) produced more accurate image enhancement results by using reinforcement learning to predict the curve parameter map at each iteration step, with users able to adjust the number of iterations.

---

[*] Equal contribution.
[†] Corresponding author.

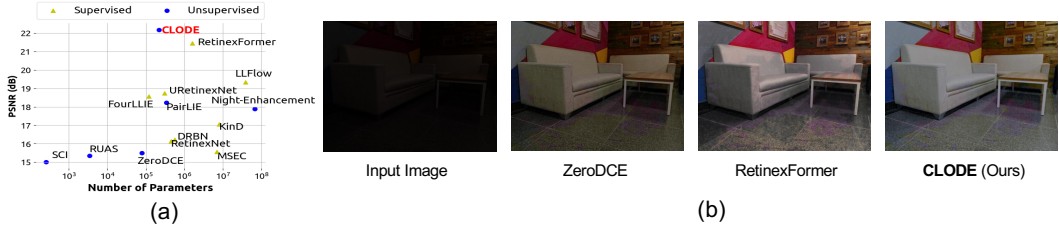

Figure 1: **(a)** Quantitative Evaluation: The average PSNR values on the LSRW (Hai et al., 2023) and LOL (Chen Wei, 2018), together with the respective parameter numbers for each model. **(b)** Visual Comparisons with ZeroDCE (Guo et al., 2020) (*unsupervised*), RetinexFormer (Cai et al., 2023) (*supervised*) and proposed CLODE (*unsupervised*).

In general, these curve-adjustment-based methods, which have fewer parameters, offer the advantage of fast and efficient training and also demonstrate the effectiveness of using higher-order curves for low-light image adjustment. However, conventional iterative approaches in discrete-space with fixed update steps do not arrive at the optimal solution and cannot guarantee convergence of the optimization. Therefore, we alleviate this problem in the discrete-space updating process of existing methods. In doing so, we bring out the strengths of curve fitting methods by reformulating the iterative update formula into ordinary differential equations. This allows the iterative approach to be transformed from discrete-space to continuous-space and to find input-specific higher-order curves until convergence within a specified tolerance. To be specific, we present the Neural Ordinary Differential Equations (NODE) model for the low-light enhancement task for the first time. By solving the NODE problem using conventional ODE solvers, we obtain better approximate solutions to the curve-adjustment problem. The proposing method therefore produces more accurate results than conventional results from iterative updates in discrete-space by exploring the continuous exposure dynamics. In this work, we introduce **C**ontinuous exposure learning for **L**ow-light image enhancement using neural **O**rdinary **D**ifferential **E**quations (**CLODE**), which is the first dynamic system for low-light image enhancement. Our main contributions can be summarized as follows:

- CLODE is the first approach to formulate the higher-order curve estimation problem as a NODE problem, enabling effective and accurate solutions with standard ODE solvers.
- By transforming the discrete update formula into NODE, which is solvable in continuous-space, we significantly enhance the unsupervised low-light image enhancement results across various benchmark datasets as shown in Fig. 1. This effectively bridges the performance gap between supervised and unsupervised approaches.
- CLODE also offers user controllability without altering the network architecture, enabling users to manually adjust the desired level of exposure as needed.

## 2 RELATED WORKS

### 2.1 UNSUPERVISED LOW-LIGHT IMAGE ENHANCEMENT

Obtaining well-exposed ground-truth images paired with corresponding low-light images is inherently challenging, which limits the use of supervised learning in low-light image enhancement. To address this limitation, many unsupervised methods have been developed to tackle the problem. First, there are some approaches (Liu et al., 2021; Ma et al., 2022; Zhao et al., 2021; Fu et al., 2023) that utilize the principles of retinex-theory. Among them, PairLIE (Fu et al., 2023) utilizes retinex-theory to identify the reflectance and illumination, and employs gamma correction with user-defined gamma values to enhance the illumination. In addition, UDCN (Jiang et al., 2022) and HEP (Zhang et al., 2021a) use histogram equalization results as a reference for exposure enhancement. Moreover, recent approaches using GANs have shown remarkable improvements by additionally utilizing unpaired images of normal exposed (Jiang et al., 2021b; Jin et al., 2022). Lastly, there are curve-adjustment-based methods (Guo et al., 2020; Li et al., 2021; 2022; Zhang et al., 2021b) that transform images through tone mapping. These methods have advanced the curve-fitting techniques from traditional editing tools into deep learning-based approaches, enhancing images by predicting the fitting curves

pixel-by-pixel. By repeating the pixel-wise curve fitting and exposure enhancement for a fixed number of iterations in discrete-space, these approaches aim to handle locally varying exposure levels (*i.e.,* single image with both underexposed and overexposed areas) in an unsupervised manner. Our CLODE also follows this unsupervised curve-adjustment-based method and reformulates the curve-fitting problem into a neural ordinary differential equation (NODE). By solving the NODE problem using conventional ODE solvers, we increase the accuracy of curve fitting and thus significantly improve the performance of low-light image enhancement.

## 2.2 Neural Ordinary Differential Equations

An ordinary differential equation (ODE) is a fundamental concept in mathematics that describes how a function changes with respect to a single variable. It captures the relationship between a function and its derivatives, providing a powerful tool for modeling dynamic systems, such as Newton's Second Law of Motion. To effectively apply the strength of ordinary differential equations to the deep learning model, the concept of neural ordinary differential equations (NODE) is introduced in (Chen et al., 2018). The use of NODE facilitates model definition and evaluation, highlighting its effectiveness in parameter efficiency, adaptive computation, and modeling continuous data. In order to effectively capture more complicated functions, the Augmented Neural ODE (ANODE) (Dupont et al., 2019) has been introduced. Furthermore, for seamless continuous time-series modeling, Latent ODE (Rubanova et al., 2019) is proposed and recently, ClimODE (Verma et al., 2023) proposed a continuous-time NODE models for numerical weather prediction. To be specific, in the field of computer vision, the Vid-ODE approach (Park et al., 2021) has been introduced to generate continuous-time videos. NODEO (Wu et al., 2022b) has presented a versatile architecture tailored for deformable image registration, and a temporal deformation model using the capabilities of NODE has been developed in (Jiang et al., 2021a) to address the challenges associated with future prediction tasks in the context of 4D reconstruction. With advantages like continuous-space modeling, adaptive computation, and memory efficiency, NODE (Chen et al., 2018) is utilized in various deep learning tasks. However, it has not been extensively explored in the field of image restoration. While NODE-SR (Park & Kim, 2022) has been introduced to address the arbitrary scale super-resolution problem, our methodology marks the first application in image exposure enhancement. In contrast to NODE-SR (Park & Kim, 2022), which learns the continuous variation of the scaling factor for the arbitrary scale super-resolution problem, our CLODE learns the continuous variation of image exposure through curve-adjustment.

## 3 Proposed Method

### 3.1 Preliminary

In photo editing applications, the curve-adjustment method is often used to adjust the tone of input images and provides effective exposure control. While this method is useful for pixel-wise manipulation, it is not well suited for images that contain areas of extreme over- or under-exposure. Additionally, a notable drawback of this approach is its reliance on manual adjustments (*e.g.,* the number of updates) by the user for each input image. This can be time-consuming and potentially less accurate in certain scenarios. To address this problem, Yuan and Sun (Yuan & Sun, 2012) have proposed a solution that aims to mitigate the limitations of manual adjustments. They introduced an automated approach that involves estimating an image-specific *S*-shaped nonlinear tone curve (referred to as an *S*-curve) tailored to each input image. Specifically, for a given low-light image $I_0$, where each pixel value is in the range [0, 1], the *S*-curve formula for the enhanced image $I_0^{'}$ can be represented as follows:

$$I_0^{'} = I_0 + \phi_s \cdot P_\Delta(I_0) - \phi_h \cdot P_\Delta(1 - I_0), \tag{1}$$

where $\phi_s$ and $\phi_h$ represent parameters for the amount of shadow and highlight, respectively. The function $P_\Delta$ serves as an increasing function for the adjustment that manipulates the intensity of individual pixels within the input of the function.

While Eq.1 allows for adjusting the brightness of an entire image using a single global curve parameter, existing iterative curve-adjustments approaches (Guo et al., 2020; Li et al., 2021; Zhang et al., 2021b; Huang et al., 2023) operate on a pixel-wise basis of the input images. Furthermore, they introduce the necessity of higher-order curves, which enhances images by fitting higher-order curves for fixed

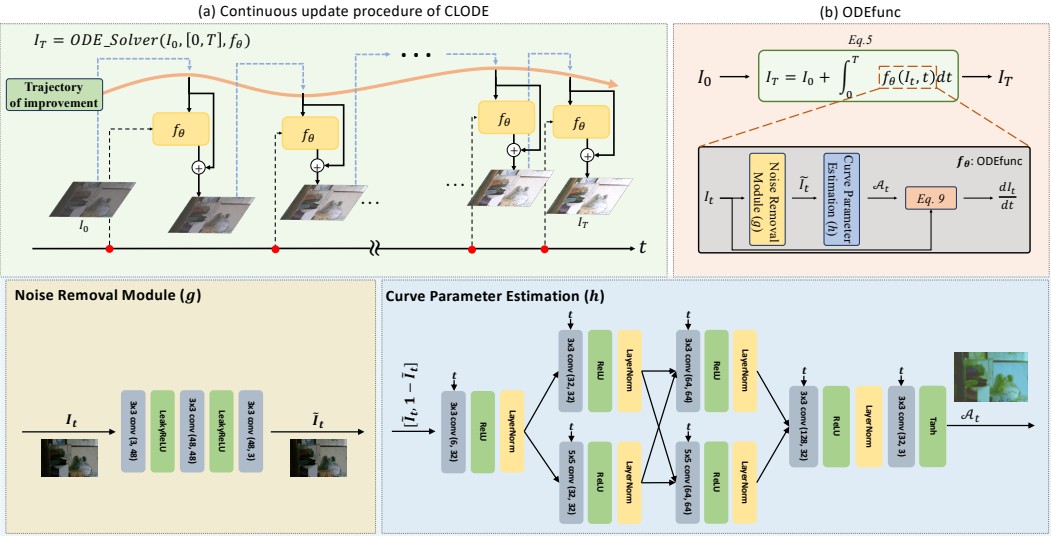

Figure 2: **(a)** Illustration of continuous update procedure of CLODE. Optimal iterative update can be achieved through the ODE equation. **(b)** Illustration of our ODEfunc $f_\theta$. ODEfunc contains the Noise Removal ($g$), Curve Parameter Estimation ($h$) module, and Eq. 9 to obtain the derivative value.

iteration steps while using a deep learning model to predict curve parameters on a pixel-by-pixel basis. Specifically the update formula enhances an image $I_n$ at the $n$-th step to an image $I_{n+1}$ at the next step as follows:

$$I_{n+1} = I_n + \mathcal{A}_n \otimes I_n \otimes (1 - I_n), \tag{2}$$

where $\mathcal{A}_n \in \mathbb{R}^{C \times H \times W}$ represents a pixel-wise varying curve parameter map and $C$, $H$, and $W$ represent the number of channels, height, and width of the image $I_n$, and $\otimes$ operation denotes element-wise multiplication. Note that, the elements of $\mathcal{A}_n$ corresponding to the curve parameters at each pixel location are in the range $[-1, 1]$ and determine the quadratic curve for the pixel-wise exposure adjustment during the enhancement process. Conventional curve-adjustment methods (Guo et al., 2020; Li et al., 2021; Zhang et al., 2021b; Li et al., 2022) iteratively follow this process for $N$ times, fitting an appropriate higher-order curve to produce the final well-exposed output image. On the contrary, our CLODE performs curve adjustment for image enhancement by reformulating Eq. 2 as an ordinary differential equation. This approach facilitates memory-efficient training and yields more accurate results through adaptive computation using modern ODE solvers.

## 3.2 CONTINUOUS EXPOSURE LEARNING FOR LOW-LIGHT IMAGE ENHANCEMENT USING NEURAL ODES

Although conventional curve-adjustment-based iterative methods offer advantages in terms of lightweight network architecture and local robustness, these approaches cannot guarantee convergence of the update process. ZeroDCE (Guo et al., 2020) empirically determines the iteration number $N$ and enhances low-light images by iterating the curve-adjustment formula 8 ($=N$) times. While ReLLIE (Zhang et al., 2021b) provides users with optional flexibility, it requires manual selection of the value of $N$ for each input image to further improve image quality. To tackle this challenge in optimization, we reformulate the curve-adjustment-based formula outlined in Eq. 2 as a Neural Ordinary Differential Equations (NODE). Then, we can solve the NODE with conventional ODE solvers (*e.g.,* Euler, RK4, dopri5) which guarantees the convergence of loss within tolerances. Specifically, we reformulate the original curve-adjustment-based formula by introducing a continuous state $t$ instead of using the discrete state $n$ as follows:

$$I_{t+1} = I_t + f_\theta(I_t, t), \tag{3}$$

where $f_\theta$ is a neural network with trainable parameters $\theta$ that satisfies $f_\theta(I_t, t) = \mathcal{A}_t \otimes I_t \otimes (1 - I_t)$. Then, we can parameterize the derivative of the enhanced image during the update using the network $f_\theta$ if the continuous update step is very small, and it is given by,

$$\frac{dI_t}{dt} = f_\theta(I_t, t). \tag{4}$$

By transforming the original curve fitting problem into a NODE problem with an initial condition $I_0$, we can estimate not only the derivative value of each state but also recover the enhanced image by solving the problem, and the initial value problem is given by,

$$I_T = I_0 + \int_0^T f_\theta(I_t, t)dt, \tag{5}$$

where $I_T$ denotes the well-exposed image at the final state $T$. Finally, the low-light image enhancement process to output $I_T$ is accomplished by using the ODE solver as:

$$I_T = \mathbf{ODE\_Solver}(I_0, [0, T], f_\theta), \tag{6}$$

where $\mathbf{ODE\_Solver}$ denotes a conventional algorithm for solving the ordinary differential equations. In our experiments, CLODE adopts the well-known dopri5 (Dormand-Prince 5th order Runge-Kutta) as an adaptive ODE solver, that determines an input-specific number of iterations for each input and dynamically adjusts the step size. Using the adaptive solver, we can adaptively compute the optimal state for different exposure levels, thereby enabling a more accurate approximation of the solution. This is in contrast to conventional methods, which use the same fixed number of iterations for all input images and cannot guarantee optimality and convergence. To the best of our knowledge, our approach is the first to define the low-light image enhancement problem as a novel NODE problem with an initial condition.

### 3.2.1 ODE FUNCTION (ODEFUNC)

We can solve the NODE problem in Eq. 5 by integrating $f_\theta$ over the time interval $[0, T]$ with the given initial value $I_0$ (*e.g.,* a low-light image). In practice, conventional ODE solvers are used to address this problem, iteratively enhancing the low-light images using Eq. 3. In Fig. 2 (a), we illustrate the continuous update procedure of our CLODE approach. Notably, the ODE function (ODEfunc) $f_\theta$ computes continuous dynamics of the latent image and is a key element in the update procedure. The detailed configuration of our ODEfunc $f_\theta$ is shown in Fig. 2 (b). To be specific, our ODEfunc includes Noise Removal ($g$) and the Curve Parameter Estimation ($h$) modules with trainable parameters, and outputs $\frac{dI_t}{dt}$, the continuous dynamics of $I_t$. Please refer to Appendix A.1.2 for more details.

**Noise Removal**  In the ODEfunc, we first employ a pre-processing step to eliminate the artifacts from $I_t$ and generate the denoised image $\tilde{I}_t$ in order to produce more accurate curve adjustment parameters $\mathcal{A}_t$. To minimize computational costs within the $f_\theta$, we employ a simple and lightweight three-layer convolutional neural network $g$ as our Noise Removal module, expressed as follows:

$$\tilde{I}_t = g(I_t). \tag{7}$$

The refined image $\tilde{I}_t$ is then used as the input to the subsequent Curve Parameter Estimation stage.

**Curve Parameter Estimation**  Inspired by (Yuan & Sun, 2012; Wang et al., 2022a), to enhance both under- and over- exposed areas, we not only use the denoised image $\tilde{I}_t$ and its inverted version $(1 - \tilde{I}_t)$ as inputs to the Curve Parameter Estimation module. The formulation is given by:

$$\mathcal{A}_t = h(\tilde{I}_t, 1 - \tilde{I}_t), \tag{8}$$

where $\mathcal{A}_t$ represents the curve parameter map at $t$, and $h$ represents the Curve Parameter Estimation module. For efficacy, this module is also a lightweight convolutional neural network. In particular, we apply layer normalization (Ba et al., 2016) to all intermediate features. Notably, the use of layer normalization enables CLODE to handle the diverse exposure ranges of input images. Furthermore, all convolutional layers within the Curve Parameter Estimation module $h$ take the continuous state $t$ as a conditional input, allowing for time-varying outputs during the integration interval $[0, T]$ as in (Chen et al., 2018).

**Continuous Dynamics**  Lastly, the derivative value of the one-step state at $t$ is computed in our ODEfunc, and it is expressed as follows:

$$\frac{dI_t}{dt} = \mathcal{A}_t \otimes I_t \otimes (1 - I_t). \tag{9}$$

Notably, unlike conventional curve-adjustment-based update formulas that discretize update steps, our continuous dynamics allows the desired level of accuracy and produces more accurate solutions.

Figure 3: **Illustration of User Controllable Design**. By manually changing the integration interval from $-(T + \Delta t)$ to $+(T + \Delta t)$, ours can produce results with different exposure levels.

### 3.3 INFERENCE PROCESS OF CLODE

**Inference Process**    Given a low-light input image $I_0$, CLODE undergoes successive image enhancement through $f_\theta$ until convergence within the specified tolerance of the ODE solvers, resulting in a well-exposed image $I_T$. Note that, the output image $I_T$ may contain some noise that is amplified during the image enhancement process. Therefore, we use the noise-free image $\tilde{I}_T$ as our final outcome by applying the Noise Removal module $g$.

**User Controllable Design**    CLODE learns the low-light exposure adjustment mechanism in the continuous-space, and is trained to output $I_T$ by integrating the states from $0$ to $T$ in Eq. 5 using a fixed $T$. However, as shown in Fig. 3, users can manually adjust the integration interval by changing the final state value $T$ at the test stage, allowing them to output images with the preferred exposure level and even produce images darker than the input. In practice, by controlling the final state from $-(T + \Delta t)$ to $(T + \Delta t)$, the exposure level of the output image can be easily controlled to provide a more user-friendly exposure level.

### 3.4 ZERO-REFERENCE LOSS FUNCTIONS

To address the challenge posed by the lack of ground truth, we use five zero-reference loss functions for unsupervised training.

**Spatial Consistency Loss**    While the given low-light input image $I_0$ is enhanced during the update procedure, maintaining spatial consistency in the pixel brightness order is crucial for preserving image details. Specifically, we measure the difference in spatial consistency between the input image $I_0$ and our prediction $I_T$ by comparing the differences in neighboring pixel values. Similar to (Guo et al., 2020), we compute the spatial consistency after applying 4-by-4 average pooling to both $I_0$ and $I_T$, and the spatial consistency loss $\mathcal{L}_{spa}$ is expressed as:

$$\mathcal{L}_{spa} = \frac{1}{K} \sum_{i=1}^{K} \sum_{j \in \Omega(i)} \left( |m_4(I_T)_i - m_4(I_T)_j| - |m_4(I_0)_i - m_4(I_0)_j| \right)^2. \tag{10}$$

The 4-by-4 average pooling operation is denoted as $m_4(\cdot)$ and $\Omega(i)$ includes neighboring pixels in four directions (left, right, top, bottom) centered at position $i$. The normalization factor $K$ denotes the number of pixels in the reduced image after the pooling operation, and $K$ is given by $\frac{H}{4} \times \frac{W}{4} \times C$.

**Exposure Loss**    To enforce a consistent exposure level across pixels, conventional unsupervised methods incorporate exposure guidance into the loss function (Guo et al., 2020). Similarly, we introduce a desired exposure level parameter E and define the exposure loss $\mathcal{L}_{exp}$ as:

$$\mathcal{L}_{exp} = ||m_{16}(I_T) - \text{E}||_2^2. \tag{11}$$

In our experiments, we set the exposure level E to 0.6, which corresponds to the gray level in the RGB color space. To maintain the overall exposure level in the results, we minimize the difference between the pixel values of the predicted image $I_T$ and the desired exposure level E after performing a 16-by-16 average pooling operation $m_{16}(\cdot)$ on the output image $I_T$.

**Color Constancy Loss**    In conventional zero-reference methods, two main approaches are used to enforce spatial color constancy: one based on the retinex-theory, and the other based on the Gray-World hypothesis in (Buchsbaum, 1980). In this work, the color constancy loss $\mathcal{L}_{col}$ is based on the Gray-World hypothesis as in (Guo et al., 2020; Zhang et al., 2021a), and the formulation is given by,

$$\mathcal{L}_{col} = (R - B)^2 + (R - G)^2 + (G - B)^2, \tag{12}$$

where $R$, $G$, and $B$ are the mean pixel values of the red, green, and blue channels in the predicted image $I_T$, respectively. We minimize the color constancy loss $\mathcal{L}_{col}$ to correct the potential color deviations in the enhanced image.

**Parameter Regularization Loss** To prevent rapid changes of pixel values in nearby regions, we employ the spatial regularization to enforce smoothness among neighboring curve parameter values in $\mathcal{A}_t$, and the formulation is given by,

$$\mathcal{L}_{param} = (|\nabla_x \mathcal{A}_0| + |\nabla_y \mathcal{A}_0|)^2 + \ldots + (|\nabla_x \mathcal{A}_{T-1}| + |\nabla_y \mathcal{A}_{T-1}|)^2, \quad (13)$$

where the linear operations $\nabla_x$ and $\nabla_y$ compute the horizontal and vertical gradients from the parameter map $\mathcal{A}_t$, respectively. For better understanding, we represent $T-1$ as the stage before the final enhancement. We employ the parameter regularization loss at each update step (*e.g.,* red points in Fig. 2 (a)) and accumulate the loss while solving the NODE problem.

**Noise Removal Loss** To estimate a spatially smooth $\mathcal{A}_t$ regardless of the noise in the image $I_t$, we use the Noise Removal module ($g$) to remove the noise. To train the Noise Removal module, we utilize a self-supervision-based loss $\mathcal{L}_{noise}$ that follows the Noise2Noise approaches (Lehtinen et al., 2018; Huang et al., 2021; Mansour & Heckel, 2023). Specifically, we employ the loss introduced in Zeroshot-N2N (Mansour & Heckel, 2023). Our $\mathcal{L}_{noise}$ has two components at state $t$: the residual loss $\mathcal{L}_{res}^t$ and the consistency loss $\mathcal{L}_{cons}^t$. We minimize these losses using two different down-samplers; $D_1$ and $D_2$. Notably, $D_1$ and $D_2$ represent fixed 2D convolutional kernels: $\begin{bmatrix} 0.5 & 0 \\ 0 & 0.5 \end{bmatrix}$ and $\begin{bmatrix} 0 & 0.5 \\ 0.5 & 0 \end{bmatrix}$, respectively. These kernels are used for downsampling through convolutions with a stride of two. First, our $\mathcal{L}_{res}^t$ fits the noise within $I_t$ through a symmetric loss function similar to the approach in (Chen & He, 2021) and it yields:

$$\mathcal{L}_{res}^t = \frac{1}{2}(||D_1(I_t) - g(D_1(I_t)) - D_2(I_t)||_2^2 + ||D_2(I_t) - g(D_2(I_t)) - D_1(I_t)||_2^2). \quad (14)$$

Next, as in (Mansour & Heckel, 2023), $\mathcal{L}_{cons}^t$ ensures spatial consistency by maintaining similarity in noise distributions, even if the order of denoising and downsampling is altered. Specifically, $\mathcal{L}_{cons}^t$ also adopts a symmetric loss and is defined as at each update step (*e.g.,* red points in Fig. 2 (a)):

$$\mathcal{L}_{cons}^t = \frac{1}{2}(||D_1(I_t) - g(D_1(I_t)) - D_1(I_t - g(I_t))||_2^2 + ||D_2(I_t) - g(D_2(I_t)) - D_2(I_t - g(I_t))||_2^2). \quad (15)$$

Therefore, our final noise removal loss $\mathcal{L}_{noise}$ can be represented accumulating during the update procedure as:

$$\mathcal{L}_{noise} = (\mathcal{L}_{res}^0 + \mathcal{L}_{cons}^0) + \ldots + (\mathcal{L}_{res}^{T-1} + \mathcal{L}_{cons}^{T-1}). \quad (16)$$

As with Eq. 13, we represent $T-1$ as the stage before the final enhancement. A more detailed description of the noise removal loss is provided in Appendix A.5.

**Final Objective Function** The final objective function to optimize is given as follows:

$$\mathcal{L}_{total} = w_{spa} \cdot \mathcal{L}_{spa} + w_{exp} \cdot \mathcal{L}_{exp} + w_{col} \cdot \mathcal{L}_{col} + w_{param} \cdot \mathcal{L}_{param} + w_{noise} \cdot \mathcal{L}_{noise}, \quad (17)$$

where $w_{spa}$, $w_{exp}$, $w_{col}$, $w_{param}$, and $w_{noise}$ are hyper-parameters used to control the relative significance of each associated loss during the training process.

## 4 EXPERIMENTS

### 4.1 IMPLEMENTATION DETAILS

Please refer to Appendix A.1 for more implementation details and training scheme.

### 4.2 EXPERIMENTAL SETUP

In this work, we use the LOL (Chen Wei, 2018) and SICE (Cai et al., 2018) Part1 datasets for training. The results of low-light image enhancement are evaluated on the LOL and LSRW (Hai et al., 2023) benchmark datasets. In addition, the SICE (Cai et al., 2018) Part2 dataset is used as a benchmark dataset for evaluation under various exposure conditions. SICE Part2 contains 229 image sequences with different exposure levels, and we use the entire sequences as the evaluation dataset. By default, each comparison model uses its official network weights. In cases where the official code is available but weights are not provided, the models are retrained using the official code and settings, except for ReLLIE (Zhang et al., 2021b). We present the performance of ReLLIE on the LOL dataset as reported in their original manuscript.

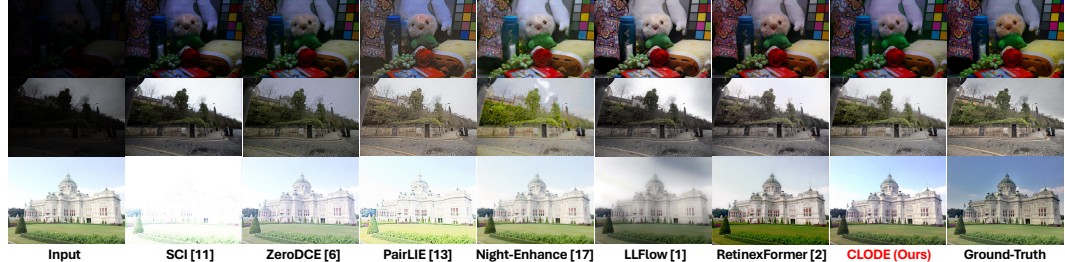

Figure 4: **Visual comparisons.** From top to bottom: LOL (Chen Wei, 2018), under- and over-exposed image of the SICE (Cai et al., 2018) Part2. For more visual results, please refer to Fig. 13 in the Appendix.

## 4.3 QUANTITATIVE COMPARISONS

First, we quantitatively compare the performance of low-light image enhancement on different datasets. Notably, in the experimental results, CLODE represents our proposed method without requiring additional user input (by default), while CLODE† represents the result of adjusting the final state $T$ to the user's preferred level, as introduced in Sec. 3.3.

In Table 1, we compare the low-light image enhancement performance on the LSRW (Hai et al., 2023) and LOL (Chen Wei, 2018) benchmark datasets in terms of peak signal-to-noise ratio (PSNR) and structural similarity (SSIM). The term "*GT Mean*" refers to the evaluation method used by KinD (Zhang et al., 2019) and LLFlow (Wang et al., 2022b), which matches the average value of the output pixels to that of the ground truth pixels. CLODE and CLODE† outperform other unsupervised learning methods. Notably, CLODE† even surpasses the PSNR of state-of-the-art supervised learning methods by 0.73 dB, when averaging the results from the LSRW and LOL datasets in the rightmost columns, without using *GT Mean*. Moreover, two notable points can be highlighted in Table 1. First, the effectiveness of using NODE to compute accurate higher order curves is evident, as demonstrated by its superiority over curve-adjustment-based methods; ZeroDCE (Guo et al., 2020) and ReLLIE (Zhang et al., 2021b). Second, unlike other models trained on the same training dataset (LOL), our model shows robust performance on both the LSRW and LOL test datasets, indicating that our model generalizes better than conventional approaches.

In Table 2, we demonstrate the robustness under various exposure conditions including both under- and over- exposures, and evaluate the performance on SICE Part2 (Cai et al., 2018). The results show that CLODE exhibits robust performance compared to other models, even under various exposure conditions. It outperforms other unsupervised learning methods, and even when compared to supervised learning methods, CLODE† and CLODE achieve the best and second best results, respectively. Despite being an unsupervised method, CLODE narrows the performance gap with state-of-the-art supervised methods. Additionally, it operates robustly under challenging conditions such as various exposure conditions in SICE Part2, surpassing supervised approaches. These strengths distinguish CLODE from other unsupervised learning methods.

## 4.4 PERCEPTUAL AND VISUAL COMPARISONS

In Table 2, we also provide a perceptual comparison of the results with other methods. The evaluation is conducted on SICE Part2, which includes a combination of underexposed and overexposed images. To measure the perceptual quality, we adopt Learned Perceptual Image Patch Similarity (LPIPS) (Zhang et al., 2018), and non-reference metrics; natural image quality evaluator (NIQE) (Mittal et al., 2012b), blind/referenceless image spatial quality evaluator (BRISQUE) (Mittal et al., 2012a), perception index (PI) (Blau et al., 2018), and Entropy (Chen et al., 2019). In these four aspects, both CLODE and CLODE† show outstanding performance compared to existing unsupervised methods. The visual results are compared in Fig. 4. CLODE shows robust and natural image enhancement results compared to other comparison methods, regardless of the exposure conditions of the input image.

Table 1: **Quantitative results on LSRW (Hai et al., 2023) and LOL (Chen Wei, 2018) datasets**. For a fair comparison, we re-trained some models on LOL and marked them with *. Among the unsupervised approaches, the best score is displayed in **red**, the second best in **blue**, and the third best in **black**. For more comparison results in terms of non-reference metrics, please refer to Appendix A.5.4.

| Training | Method | #Params (M) | Train dataset | LSRW | | | | LOL | | | | Average | | | |
|---|---|---|---|---|---|---|---|---|---|---|---|---|---|---|---|
| | | | | Normal | | GT Mean | | Normal | | GT Mean | | Normal | | GT Mean | |
| | | | | PSNR↑ | SSIM↑ | PSNR↑ | SSIM↑ | PSNR↑ | SSIM↑ | PSNR↑ | SSIM↑ | PSNR↑ | SSIM↑ | PSNR↑ | SSIM↑ |
| Supervised | RetinexNet (Chen Wei, 2018) | 0.4446 | LOL | 15.49 | 0.355 | 16.55 | 0.371 | 16.77 | 0.419 | 17.65 | 0.648 | 16.13 | 0.387 | 17.10 | 0.510 |
| | URetinexNet (Wu et al., 2022a) | 0.3069 | LOL, SICE | 17.63 | 0.516 | 18.10 | 0.523 | 19.84 | 0.826 | 21.33 | 0.835 | 18.74 | 0.671 | 19.71 | 0.679 |
| | DRBN (Yang et al., 2021) | 0.5556 | LOL | 16.15 | 0.542 | 17.68 | 0.548 | 16.29 | 0.617 | 19.55 | 0.746 | 16.22 | 0.580 | 18.62 | 0.647 |
| | KinD (Zhang et al., 2019) | 8.0160 | LOL | 16.47 | 0.493 | 19.86 | 0.504 | 17.65 | 0.775 | 20.87 | 0.802 | 17.06 | 0.634 | 20.36 | 0.653 |
| | LLFlow (Wang et al., 2022b) | 38.859 | LOL | 17.52 | 0.509 | 18.68 | 0.518 | 21.15 | 0.854 | 24.99 | 0.871 | 19.34 | 0.681 | 21.84 | 0.694 |
| | RetinexFormer (Cai et al., 2023) | 1.6057 | LOL | 17.76 | 0.517 | 19.15 | 0.529 | 25.15 | 0.845 | 27.18 | 0.850 | 21.45 | 0.681 | 23.17 | 0.690 |
| | GSAD (Hou et al., 2023) | 17.434 | LOL | 17.37 | 0.509 | 19.51 | 0.525 | 23.01 | 0.850 | 27.60 | 0.875 | 20.19 | 0.680 | 23.55 | 0.700 |
| | PyDiff (Zhou et al., 2023) | 97.886 | LOL | 17.00 | 0.516 | 20.11 | 0.545 | 20.49 | 0.855 | 26.99 | 0.882 | 18.75 | 0.686 | 23.55 | 0.713 |
| Unsupervised | SCI-easy (Ma et al., 2022) | 0.0003 | MIT-5K | 11.79 | 0.317 | 16.97 | 0.426 | 9.58 | 0.369 | 18.55 | 0.501 | 10.69 | 0.343 | 17.76 | 0.464 |
| | SCI-medium (Ma et al., 2022) | 0.0003 | LOL, LSRW | 15.24 | 0.424 | 17.84 | 0.439 | 14.78 | 0.521 | 19.11 | 0.504 | 15.01 | 0.473 | 18.47 | 0.472 |
| | SCI-difficult (Ma et al., 2022) | 0.0003 | DARKFace | 15.16 | 0.408 | 18.04 | 0.424 | 13.81 | 0.526 | 19.64 | 0.510 | 14.48 | 0.467 | 18.84 | 0.467 |
| | SCI* (Ma et al., 2022) | 0.0003 | LOL | 14.82 | 0.413 | 17.65 | 0.437 | 13.84 | 0.507 | 19.02 | 0.499 | 14.33 | 0.460 | 18.34 | 0.468 |
| | RUAS* (Liu et al., 2021) | 0.0034 | LOL | 14.27 | 0.470 | 17.10 | 0.509 | 16.41 | 0.500 | 18.65 | 0.520 | 15.34 | 0.485 | 17.88 | 0.514 |
| | ZeroDCE* (Guo et al., 2020) | 0.0794 | LOL | 14.50 | 0.403 | 18.87 | 0.467 | 16.49 | 0.522 | 20.99 | 0.596 | 15.50 | 0.463 | 19.93 | 0.532 |
| | ReLLIE (Zhang et al., 2021b) | - | - | - | - | - | - | 18.37 | 0.641 | - | - | - | - | - | - |
| | PairLIE (Fu et al., 2023) | 0.3417 | LOL, SICE | 16.97 | 0.498 | 18.82 | 0.523 | 19.51 | 0.736 | 23.10 | 0.752 | 18.24 | 0.617 | 20.96 | 0.637 |
| | Night-Enhancement (Jin et al., 2022) | 67.011 | LOL | 14.24 | 0.472 | 19.19 | 0.554 | 21.52 | 0.763 | 24.25 | 0.781 | 17.88 | 0.618 | 21.72 | 0.668 |
| | **CLODE** | 0.2167 | LOL | 17.28 | 0.533 | 20.60 | 0.557 | 19.61 | 0.718 | 23.16 | 0.752 | 18.44 | 0.625 | 21.88 | 0.655 |
| | **CLODE** † | 0.2167 | LOL | 20.77 | 0.562 | 20.94 | 0.568 | 23.58 | 0.754 | 24.47 | 0.759 | 22.18 | 0.658 | 22.71 | 0.664 |

Table 2: **Quantitative results on SICE (Cai et al., 2018) Part2**. For a fair comparison, we re-trained some models on SICE Part 1 and marked them with *. Within the unsupervised approaches, the best score is displayed in **red**, the second in **blue** and the third in **black**.

| Training | Method | Train dataset | Normal | | | | | | | GT Mean | |
|---|---|---|---|---|---|---|---|---|---|---|---|
| | | | PSNR↑ | SSIM↑ | LPIPS↓ | NIQE↓ | BRISQUE↓ | PI↓ | Entropy↑ | PSNR↑ | SSIM↑ |
| Supervised | URetinexNet (Wu et al., 2022a) | LOL, SICE | 12.15 | 0.708 | 0.393 | 4.250 | 15.633 | 3.372 | 6.926 | 17.81 | 0.686 |
| | LLFlow* (Wang et al., 2022b) | SICE | 14.34 | 0.608 | 0.279 | 3.643 | 17.011 | 3.481 | 6.566 | 19.59 | 0.658 |
| | ECLNet (Huang et al., 2022b) | SICE | 13.99 | 0.562 | 0.290 | 4.279 | 24.570 | 3.520 | 6.919 | 16.66 | 0.690 |
| | FECNet (Huang et al., 2022a) | SICE | 14.25 | 0.600 | 0.291 | 3.786 | 17.454 | 3.025 | 7.035 | 16.47 | 0.639 |
| | RetinexFormer* (Cai et al., 2023) | SICE | 19.12 | 0.570 | 0.369 | 4.452 | 24.768 | 4.573 | 7.025 | 20.97 | 0.578 |
| | RetinexFormer (Cai et al., 2023) | MIT-5K | 13.23 | 0.564 | 0.263 | 3.848 | 17.350 | 2.863 | 6.881 | 16.35 | 0.609 |
| Unsupervised | SCI-easy (Ma et al., 2022) | MIT-5K | 9.87 | 0.486 | 0.372 | 4.276 | 21.850 | 3.226 | 6.113 | 16.44 | 0.622 |
| | SCI-medium (Ma et al., 2022) | LOL, LSRW | 9.77 | 0.510 | 0.454 | 5.727 | 33.200 | 4.392 | 5.212 | 15.83 | 0.574 |
| | SCI-difficult (Ma et al., 2022) | DarkFace | 11.13 | 0.577 | 0.324 | 4.636 | 23.620 | 3.107 | 6.386 | 16.85 | 0.647 |
| | SCI* (Ma et al., 2022) | SICE | 10.67 | 0.478 | 0.331 | 4.289 | 23.449 | 3.570 | 6.213 | 17.99 | 0.675 |
| | RUAS* (Liu et al., 2021) | SICE | 9.12 | 0.408 | 0.539 | 8.097 | 52.923 | 6.004 | 5.101 | 15.52 | 0.531 |
| | ZeroDCE (Guo et al., 2020) | SICE | 12.67 | 0.635 | 0.244 | 3.886 | 21.630 | 2.821 | 6.516 | 18.85 | 0.686 |
| | PairLIE (Fu et al., 2023) | LOL, SICE | 13.39 | 0.619 | 0.305 | 5.268 | 36.536 | 3.548 | 6.376 | 19.22 | 0.663 |
| | Night-Enhancement* (Jin et al., 2022) | SICE | 13.18 | 0.581 | 0.360 | 4.728 | 33.883 | 4.133 | 6.661 | 19.43 | 0.660 |
| | **CLODE** | SICE | 15.01 | 0.687 | 0.239 | 4.050 | 18.663 | 3.005 | 7.006 | 19.64 | 0.706 |
| | **CLODE†** | SICE | 16.18 | 0.707 | 0.200 | 4.026 | 18.210 | 2.970 | 7.045 | 21.55 | 0.813 |

## 4.5 COMPARISON ON COLOR CASTS

CLODE enhances the image in an unsupervised manner based on the color statistics of the input image, which can lead to color casts. Nevertheless, CLODE exhibits less color casts compared to previous unsupervised methods.

In Table 3, we measure the degree of color casts between the output image and the ground-truth image using the color-matching histogram loss (Afifi et al., 2021a), which is designed to control the color of the input image by matching color histogram with the target image. Additionally, we present the results of color correction performance in the LAB color space using $\Delta E_{2000}$ and $\Delta E_{ab}$, well-established color metrics. For all metrics used in the evaluation, lower values indicate a closer match to the colors of the ground-truth image. As shown in Table 3, CLODE using the NODE scenario demonstrates superior performance in terms of naturalness, image quality metrics, and color-matching histogram loss compared to existing methods. Moreover, the results of $\Delta E_{2000}$ and $\Delta E_{ab}$ confirm that CLODE† achieves the best performance, and CLODE ranks second, implying that our method leads to less color cast problem and offers superior enhancement effects.

Table 3: Quantitative comparisons on LOL (Chen Wei, 2018)/SICE (Cai et al., 2018) dataset.

| Method | NIQE↓ | BRISQUE↓ | color-matching histogram loss↓ | $\Delta E_{2000}$↓ (Sharma et al., 2005) | $\Delta E_{ab}$↓ (Sharma & Bala, 2017) |
|---|---|---|---|---|---|
| SCI-easy (Ma et al., 2022) | 7.15/4.28 | 12.42/21.85 | 0.4860/0.4788 | 31.49/27.00 | 39.21/35.13 |
| SCI-medium (Ma et al., 2022) | 7.86/5.73 | 25.87/33.20 | 0.4530/0.4911 | 19.40/27.28 | 27.28/35.96 |
| SCI-difficult (Ma et al., 2022) | 8.06/4.64 | 26.82/23.62 | 0.3854/0.4872 | 21.06/24.02 | 26.09/31.05 |
| RUAS (Liu et al., 2021) | 6.30/8.10 | 11.98/52.92 | 0.4471/0.5100 | 16.80/29.18 | 29.18/38.83 |
| Zero-DCE (Guo et al., 2020) | 7.78/3.89 | 27.30/21.63 | 0.4485/0.4647 | 21.93/21.26 | 26.60/27.10 |
| CLODE | 4.52/4.05 | 8.22/18.66 | 0.4381/0.4606 | 12.73/17.04 | 15.71/22.39 |
| CLODE† | 4.25/4.03 | 8.81/18.21 | 0.3848/0.4462 | 9.21/14.46 | 11.86/19.32 |

## 4.6 ABLATION STUDY

**Effectiveness of NODE** To validate the impact of NODE, we compare curve adjustment in discrete (w/o NODE) and continuous (w/ NODE) spaces using CLODE's architecture, as shown in Table 4.

Table 4: Comparative experiments according to using NODE on LSRW (Hai et al., 2023)/LOL (Chen Wei, 2018).The "Discrete" refers to performing curve adjustment in discrete steps, similar to the conventional methods (Guo et al., 2020; Zhang et al., 2021b), and "Continuous" refers to the reformulation of NODE.

| Method | Case | Step ($N$) | PSNR↑ | SSIM↑ | BRISQUE↓ |
|---|---|---|---|---|---|
| Discrete | (a1) | 1 | 11.19/9.236 | 0.297/0.362 | 41.137/41.169 |
| | (b1) | 5 | 16.12/17.47 | 0.419/0.716 | 31.421/33.042 |
| | (c1) | 10 | 13.94/16.18 | 0.395/0.520 | 32.267/32.243 |
| | (d1) | 20 | 12.95/14.94 | 0.373/0.506 | 33.537/34.941 |
| | (e1) | 30 | 12.87/14.97 | 0.375/0.509 | 33.537/35.342 |
| Continuous | (f1) | ≤ 30 (adaptive) | **17.28/19.61** | **0.533/0.718** | **18.426/8.220** |

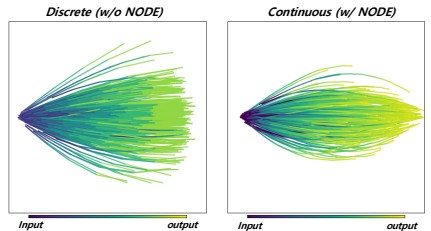

Figure 5: Trajectories of improvement for (e1) and (f1) in Table 4. PCA dimension reduction is used to visualize the trajectories.

Table 5: Impact of the modules in $f_\theta$. Noise Removal and the layer normalization (LN) significantly improve performance.

| Case | Noise Removal $g$ | LN in $h$ | PSNR↑ | SSIM↑ |
|---|---|---|---|---|
| (a2) | | | 14.72 | 0.538 |
| (b2) | ✓ | | 15.19 | 0.489 |
| (c2) | | ✓ | 18.67 | 0.577 |
| (d2) | ✓ | ✓ | 19.61 | 0.718 |

Table 6: Execution time and performance.

| Training | Method | PSNR/SSIM | #Params (M) | Time (S) |
|---|---|---|---|---|
| Supervised | RetinexNet (Chen Wei, 2018) | 15.49/0.355 | 0.4446 | 0.337 |
| | LLFlow (Wang et al., 2022b) | 17.52/0.509 | 38.859 | 0.144 |
| | RetinexFormer (Cai et al., 2023) | 17.76/0.517 | 1.6057 | 0.072 |
| Unsupervised | SCI-medium (Ma et al., 2022) | 15.24/0.424 | 0.0003 | 0.001 |
| | RUAS (Liu et al., 2021) | 14.27/0.470 | 0.0034 | 0.006 |
| | ZeroDCE (Guo et al., 2020) | 15.81/0.449 | 0.0794 | 0.004 |
| | PairLIE (Fu et al., 2023) | 16.97/0.498 | 0.3417 | 0.008 |
| | **CLODE** | 17.28/0.533 | 0.2167 | 0.056 |
| | **CLODE-S** | 16.97/0.457 | 0.0004 | 0.005 |

In the discrete setting, similar to (Guo et al., 2020), curve parameters are estimated in parallel for fixed steps [1, 5, 10, 20, 30] ((a1)–(e1)), while in the continuous setting, parameters are sequentially estimated for adaptive steps, up to 30 ((f1)). Table 4 shows that the sequential continuous updates produce more accurate parameters, demonstrating superior performance over the conventional discrete approach. Additionally, Fig. 5 visualizes latent image trajectories during updates using PCA, revealing that continuous adjustments ((f1)) converge more accurately than discrete updates ((e1)), highlighting NODE's contribution to image enhancement. For visualization result on each case and further detailed explanation on NODE, please refer to Sec. A.1.1 and Fig. 6 of the Appendix.

**Effect of the Modules**  In Table 5, we conduct ablation experiments on the modules used in ODEfunc $f_\theta$. We verify the effects of the Noise Removal module $g$ and the layer normalization (LN) in the Curve Parameter Estimation module $h$. Each module shows performance improvements compared to the baseline (a2). In particular, our final model (d2) achieves the largest performance gain in terms of PSNR/SSIM. Furthermore, case (c2), which includes layer normalization, has about a 4dB gain in PSNR compared to (a2), which does not include layer normalization. This shows that during the image enhancement process in NODE, it is essential to use layer normalization to normalize each state. The visual results can be seen in Fig. 9 of the Appendix.

## 5  LIMITATIONS

Table 6 presents the PSNR/SSIM performance, parameter count, and execution time measured on LSRW (Hai et al., 2023) using an NVIDIA RTX 4090. CLODE demonstrates a size advantage over supervised methods. While its iterative ODE solving takes longer than lightweight unsupervised models, it achieves comparable speed and performance to supervised approaches. Additionally, a smaller variant, CLODE-S (Appendix A.1.2), shows promising enhancement with inference times similar to unsupervised models.

## 6  CONCLUSIONS

In this work, we address the unsupervised low-light image enhancement problem by reframing discrete iterative curve-adjustment methods into a continuous space using Neural Ordinary Differential Equations (NODE). Our proposed CLODE method effectively overcomes the limitations of existing approaches, demonstrating superior convergence and adaptability in diverse low-light and multi-exposure scenarios. Additionally, CLODE introduces user-controllability, leveraging NODE's inherent flexibility to deliver customizable brightness adjustments. By incorporating a novel curve-adjustment framework and offering enhanced interpretability, our method bridges the gap between unsupervised and supervised approaches, representing a meaningful contribution to the field.

ACKNOWLEDGMENTS

This work was supported by Samsung Electronics Co., Ltd, and Samsung Research Funding Center of Samsung Electronics under Project Number SRFCIT1901-06, and Institute of Information & communications Technology Planning & Evaluation (IITP) grant funded by the Korea government (MSIT) (No.2022- 0-00156, Fundamental research on continual meta-learning for quality enhancement of casual videos and their 3D metaverse transformation).

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

## A APPENDIX

### A.1 IMPLEMENT DETAILS

The training set of images is resized to 128x128, we employ the *Pytorch* framework on NVIDIA A6000 GPU with a batch size of 8. The ADAM optimizer is used with default parameters and a fixed learning rate of $1e^{-5}$ to optimize the parameters of our network. The weights for the loss function $w_{col}$, $w_{param}$, $w_{spa}$, $w_{exp}$ and $w_{noise}$ are set to 20, 200, 1, 10 and 1 respectively, to balance the scale of losses. Furthermore, we adopt *torchdiffeq* (Chen, 2018) for Neural ODEs implementation. The training process is conducted for 100 epochs.

### A.1.1 IMPLEMENTATION DETAILS OF NODE

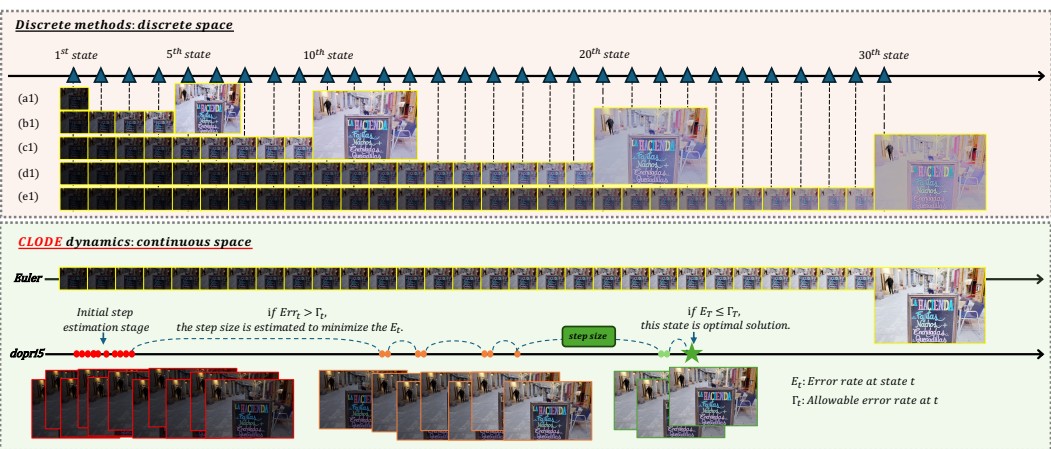

Figure 6: Further analysis of CLODE and previous discrete methods as described in Table 4 of main manuscript.

As mentioned in NODE (Chen et al., 2018), ODE reformulation provides benefits such as continuous space estimation, memory efficiency, and accurate problem-solving with ODE solvers. In Fig. 6, we describe additional analysis of CLODE. The top of Fig. 6 shows discrete trajectories of models (a1) to (e1) from Table 4, while the bottom shows CLODE trajectories with Euler and dopri5 (corresponding to Table 4. (f1)) solvers. (Top) Discrete methods (a1) to (e1) enhance images but don't achieve optimal exposure. (Bottom) CLODE (dopri5) provides more realistic image enhancement in continuous space.

Additionally, the early stop mechanism of the adaptive solver is explained at the bottom of Fig. 6. CLODE (dopri5) uses an early stop mechanism. It tracks error at each state, terminating when the error is within allowable error rate. For dopri5, $k$-order solutions ($k$=5) are used to calculate error ($\Gamma_t$) as follows:

$$\Gamma_t = atol + rtol \times norm(|O_t^K - O_t^{K-1}|), \tag{18}$$

where the $k$-order solution at time $t$ is denoted as $O^K$ and the $(k-1)$-order solution is denoted as $O_t^{K-1}$. $atol$ is absolute tolerance, and $rtol$ is relative tolerate, and the norm being used is a mixed L-infinity/RMS norm.

If $|O_t^K - O_t^{K-1}| > \Gamma_t$ the step size is re-adjusted, or it's within $\Gamma_t$, the solution is deemed optimal, and the process terminates. ODE solvers are designed to find optimal solutions through iterative steps. The top part of Fig. 6 shows that discrete methods can't guarantee optimal solutions, which led us to develop the NODE method for continuous ODE problems. Thus, improvements are due more to NODE reformulation than to iteration count. Table 4 shows that NODE outperforms simple discrete repetition. For example, using the Euler method in 30 steps achieves better performance than method (The visual result in Fig. 6 (e1) is inferior to CLODE with Euler applied.). We chose dopri5 (Dormand-Prince Runge-Kutta of Order 5) as CLODE's default solver for its stability and reliability across platforms like MATLAB. The maximum allowed step for the adaptive solver is set to 30. In

Eq. 18, the relative and absolute tolerances for the error rate calculation are set uniformly to $1e^{-5}$. We set both $atol$ and $rtol$ to 1e-5.

### A.1.2 DETAILS OF THE CLODE ARCHITECTURE

This section presents the architectural details of the CLODE network architecture, with a particular focus on the ODEfunc module. The Noise Removal module $g$ employs a simple and lightweight three-layer convolutional network. In Curve Parameter Estimation module $h$, a shallow network with two branches is utilized, wherein filters of varying sizes are employed at each branch to capture image features across different filter scales. We also provide architectural details of CLODE-S as mentioned in Sec. 5 of the main manuscript. This version omits the Noise Removal module for speed and uses a 2-layer network with 1x1 convolutions.

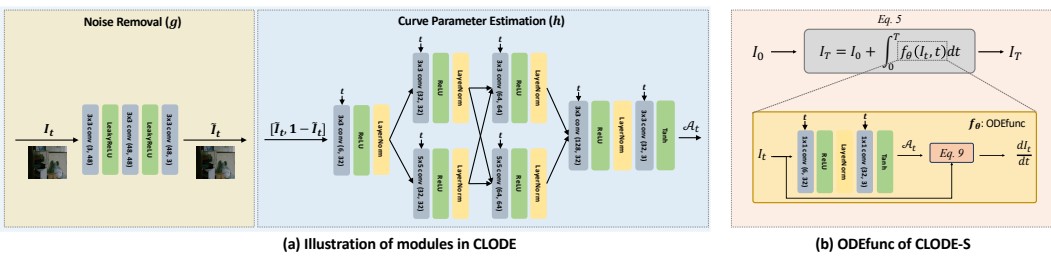

Figure 7: Illustration of architecture details of (a) modules of ODEfunc in CLODE and (b) ODEfunc of CLODE-S.

## A.2 IMPACT OF EACH LOSS FUNCTIONS

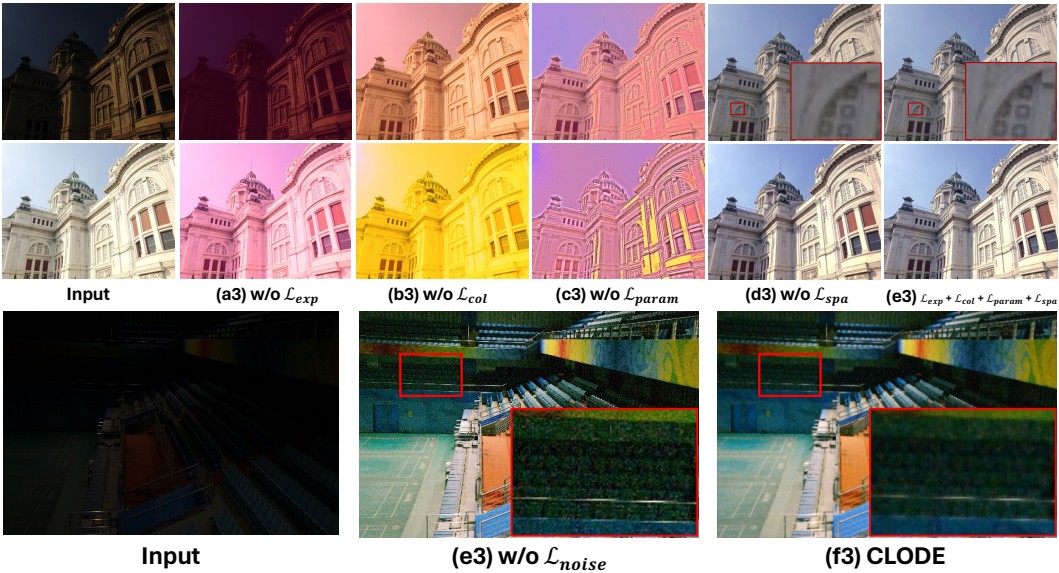

Figure 8: Visual results for the ablation study of each loss function. CLODE combines five non-reference loss functions in training for producing optimal enhancement results.

CLODE combines five non-reference loss functions to train NODE, producing optimal improvements. We present ablation experiments for each loss function, and the results are presented in Table 7 and Fig. 8. The results of each image ablation experiment demonstrate that appropriate improvement results can only be obtained when using CLODE with all loss functions. The characteristics of the loss function as observed in each ablation are as follows: **((a3) w/o $\mathcal{L}_{exp}$)**: Brightness improvement

Table 7: Ablation study on each non-reference losses. The experiment is evaluated on LOL (Chen Wei, 2018).

| Case | $\mathcal{L}_{spa}$ | $\mathcal{L}_{exp}$ | $\mathcal{L}_{col}$ | $\mathcal{L}_{param}$ | $\mathcal{L}_{noise}$ | PSNR | SSIM |
|------|------|------|------|------|------|------|------|
| (a3) | ✓ | | ✓ | ✓ | | 8.84 | 0.323 |
| (b3) | ✓ | ✓ | | ✓ | | 14.72 | 0.566 |
| (c3) | ✓ | ✓ | ✓ | | | 14.76 | 0.535 |
| (d3) | | ✓ | ✓ | ✓ | | 18.76 | 0.580 |
| (e3) | ✓ | ✓ | ✓ | ✓ | | 18.92 | 0.582 |
| (f3) | ✓ | ✓ | ✓ | ✓ | ✓ | **19.61** | **0.718** |

is not achieved in low-exposure enhancement. **(b3) w/o $\mathcal{L}_{col}$**): Severe color distortion occurs in over-exposure enhancement, damaging structural details. **(c3) w/o $\mathcal{L}_{param}$**): Structural distortion occurs, creating artifacts. **(d3) w/o $\mathcal{L}_{spa}$**): While showing better results than other experiments, it occurs loss of structural details compared to (e3). **(e3) w/o $\mathcal{L}_{Noise}$**): Compared to the proposed version (f3), it produces improved results with noise present.

## A.3 VISUALIZATION OF CURVE PARAMETER MAP $\mathcal{A}$

We provide visual comparison results for the module ablation experiments in Sec. 4.6 of the main manuscript. In the visual results without Noise Removal module (c2), we can observe the noise in $\mathcal{A}$. The enhanced result of (c2) using $\mathcal{A}$ with noise shows overall color discrepancy compared to the ground-truth, in contrast to the enhanced result of (d2) where the Noise Removal module are applied. The enhanced result of (d2) shows robust color similarity with the ground-truth image. We can confirm that removing noise for $\mathcal{A}$ is important for curve-adjustment-based method.

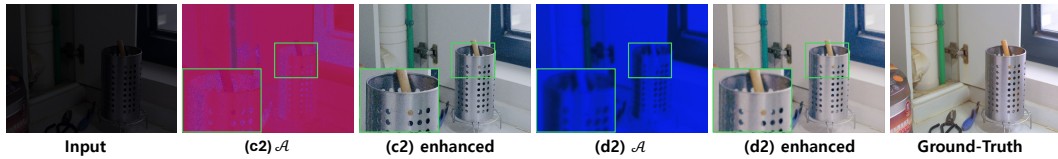

| Input | (c2) $\mathcal{A}$ | (c2) enhanced | (d2) $\mathcal{A}$ | (d2) enhanced | Ground-Truth |

Figure 9: A visual comparison of the results for (c2) and (d2) from Table 5 in the main manuscript. The enhanced result (d2) using $\mathcal{A}$ with noise removal module demonstrates improvement more similar to the ground-truth.

## A.4 CLODE STEP STATISTICS ACROSS EXPOSURE CONDITIONS.

In CLODE, the maximum allowable step for the ODE solver is empirically set to 30, considering speed. The ODE solver terminates early if it finds the optimal solution within the maximum steps. Furthermore, in order to provide the statistical analysis, the average number of steps for the ODE solver was calculated across the SICE, BSD100, DIV2K, and LOL datasets. The SICE dataset comprises five to seven images per sample, with exposure levels ranging from under-exposed to over-exposed. Additionally, BSD100 and DIV2K were used to provide additional statistics for normal-exposed conditions. The results based on the exposure conditions are in Table 8.

Table 8: Step Statistics across exposure conditions

| Condition | SICE | BSD100, DIV2K | LOL |
|-----------|------|---------------|-----|
| Under-exposed | 13.716 | - | 21.120 |
| Normal-exposed | 20.754 | 28.210 | 26.400 |
| Over-exposed | 18.906 | - | - |

The number of calculation steps increases in the order of under-exposed, over-exposed, and normal-exposed images. This is because improving a normal-exposed image is considered a stiff problem.

For normal-exposed images, where minimal improvement is required, the dynamics are more stiff than for other images. Specifically, for dynamically stiff ODE problems, the step size taken by the solver is forced down to a small level even in a region where the solution curve is smooth, and these decreased step sizes may require more evaluation steps.

For clearer comprehension, we present additional graph results in Fig. 10 (Left). In the inference time, CLODE aims to find the optimal solution by minimizing the loss functions, therefore in Fig. 10 (Left), the y-axis represents the non-reference loss value, while the x-axis represents time, with each point indicating a step. The dopri5 solver, which we primarily use, is a non-stiff solver. Although a stiff solver (e.g., ode15s) could potentially reduce the number of inference steps for normally exposed images, it proves inefficient for improving under- or over-exposed images, which are closer to non-stiff problems. Future research may explore dynamically adaptive ODE solver algorithms that adjust based on input image conditions by this observation.

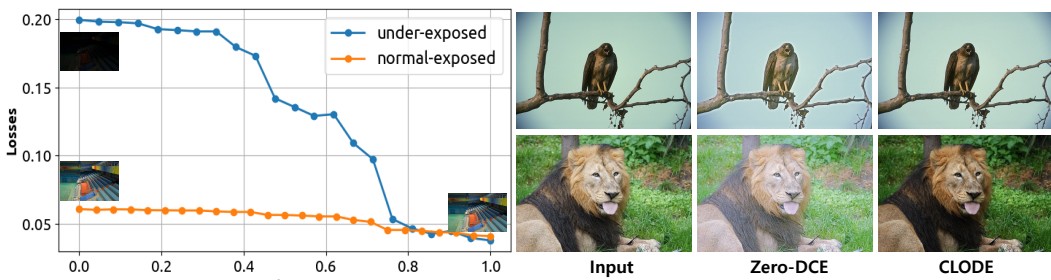

Figure 10: **(Left)** The trajectory of loss changes over time. **(Right)** Comparison of enhancement results for normal-exposed inputs.

## A.5 BACKGROUND OF NOISE REMOVAL LOSS

In Sec.3.4 we provide information about the zero-reference loss functions that we used. Unlike the others, the Noise Removal Loss ($\mathcal{L}_{noise}$) requires more explanation due to its unfamiliarity in the field of low-light enhancement, so we provide additional explanation for it.

### A.5.1 NOISE2NOISE BACKGROUND

In supervised denoising studies, neural networks are aimed at denoising the noisy image $\mathbf{y}$ to the clean image $\mathbf{x}$. Since the noisy $\mathbf{y}$ is an addition of the clean image $\mathbf{x}$ and the noise $\mathbf{e}$, the network is trained to map the noise $\mathbf{e}$ which is called Noise2Clean (N2C) method. If the network parameter is $\phi_{N2C}$, the object function of the supervised denoising method with the network $g_\phi$ can be written as:

$$\phi_{N2C} = \arg\min_{\phi} \mathbb{E}\big[||g_\phi(\mathbf{y}) - \mathbf{x}||_2^2\big]. \tag{19}$$

Denoising networks can also be trained to output the noisy image $\mathbf{y}_2$ from the noisy input image $\mathbf{y}_1$ that comes from the same clean image $\mathbf{x}$. This noise-to-noise manner can be achieved by assuming that the noise has a mean of zero as introduced in Noise2Noise (N2N) (Lehtinen et al., 2018). This is the objective function for the N2N network parameter $\phi_{N2N}$:

$$\phi_{N2N} = \arg\min_{\phi} \mathbb{E}\big[||g_\phi(\mathbf{y}_2) - \mathbf{y}_1||_2^2\big]. \tag{20}$$

The N2N manner shows close performance compare to N2C manner with sufficient training data since the objective functions of N2C and N2N are aimed on the same network parameter. If $\mathbf{y}_a = \mathbf{x} + \mathbf{e}_a$, $\mathbf{y}_b = \mathbf{x} + \mathbf{e}_b$, and the mean value of $\mathbf{e}_a$ and $\mathbf{e}_b$ are zero, the proof is as follows:

$$\phi_{N2C} = \arg\min_{\phi} \mathbb{E}\big[||g_\phi(\mathbf{y}_2) - \mathbf{x}||_2^2\big]$$

$$= \arg\min_{\phi} \mathbb{E}\big[||g_\phi(\mathbf{y}_2)||_2^2 - 2\mathbf{x}^\mathsf{T} g_\phi(\mathbf{y}_2) + ||\mathbf{x}||_2^2\big]$$

$$= \arg\min_{\phi} \mathbb{E}\big[||g_\phi(\mathbf{y}_2)||_2^2 - 2\mathbf{x}^\mathsf{T} g_\phi(\mathbf{y}_2)\big]$$

$$\phi_{N2N} = \arg\min_{\phi} \mathbb{E}\big[||g_\phi(\mathbf{y}_2) - \mathbf{y}_1||_2^2\big]$$

$$= \arg\min_{\phi} \mathbb{E}\big[||g_\phi(\mathbf{y}_2) - (\mathbf{x} + \mathbf{e}_1)||_2^2\big] \qquad (21)$$

$$= \arg\min_{\phi} \mathbb{E}\big[||g_\phi(\mathbf{y}_2)||_2^2 - 2\mathbf{x}^\mathsf{T} g_\phi(\mathbf{y}_2) - 2\mathbf{e}_2^\mathsf{T} g_\phi(\mathbf{y}_2) + ||\mathbf{x} + \mathbf{e}_1||_2^2\big]$$

$$= \arg\min_{\phi} \mathbb{E}\big[||g_\phi(\mathbf{y}_2)||_2^2 - 2\mathbf{x}^\mathsf{T} g_\phi(\mathbf{y}_2) - 2\mathbf{e}_2^\mathsf{T} g_\phi(\mathbf{y}_2)\big]$$

$$= \arg\min_{\phi} \mathbb{E}\big[||g_\phi(\mathbf{y}_2)||_2^2 - 2\mathbf{x}^\mathsf{T} g_\phi(\mathbf{y}_2)\big].$$

By Eq. 21 we can confirm that the object of $\phi_{N2C}$ and $\phi_{N2N}$ is the identical one.

### A.5.2 ZEROSHOT NOISE2NOISE METHOD

In spite of N2N approaches, it is hard to obtain two different noisy images from the same clean scene. To address this hurdle, the Neighbor2Neighbor (Huang et al., 2021) method is proposed. This allows a pair of noisy images to be augmented from a single noisy image coming from the same clean image. In Zeroshot-N2N (Mansour & Heckel, 2023), which is adopted in our proposed method, Neighbor2Neighbor is achieved by using two different 2D convolutional kernels ($D_1$ and $D_2$) on noisy images. If the noisy image is $\mathbf{y}$, a pair of down-sampled images $\mathbf{y}_1$, $\mathbf{y}_2$ can be represented as:

$$\mathbf{y}_1 = D_1(\mathbf{y}), \mathbf{y}_2 = D_2(\mathbf{y}). \qquad (22)$$

For a noisy image $\mathbf{y}$ with a size of $H \times W \times C$, the size of $\mathbf{y}_1$ and $\mathbf{y}_2$ is $\frac{H}{2} \times \frac{W}{2} \times C$. With downsampled images $\mathbf{y}_1$ and $\mathbf{y}_2$, the loss optimizes $g_\phi$ to fit the noise as:

$$\arg\min_{\phi} ||g_\phi(\mathbf{y}_1) - \mathbf{y}_2||_2^2. \qquad (23)$$

Zeroshot-N2N (Mansour & Heckel, 2023) emphasizes that residual learning, a symmetry loss, and an additional coherence-enhancing term are critical for good performance. Zeroshot-N2N proposes two different loss functions, the residual loss $\mathcal{L}_{res}$ and the consistency loss $\mathcal{L}_{cons}$. First, the residual loss optimizes the network $g_\phi$ to fit the noise instead of image. The loss then becomes as:

$$\arg\min_{\phi} ||\mathbf{y}_1 - g_\phi(\mathbf{y}_1) - \mathbf{y}_2||_2^2. \qquad (24)$$

To fit the noise in $\mathbf{y}_1$ and $\mathbf{y}_2$ both, a symmetric loss (Chen & He, 2021) is applied as:

$$\mathcal{L}_{res}(\phi) = \frac{1}{2}\big(||\mathbf{y}_1 - g_\phi(\mathbf{y}_1) - \mathbf{y}_2||_2^2 + ||\mathbf{y}_2 - g_\phi(\mathbf{y}_2) - \mathbf{y}_1||_2^2\big). \qquad (25)$$

Second, the method constrain consistency by making denoised output of the downsampled image and downsampled result of the denoised image like:

$$\arg\min_{\phi} ||\mathbf{y}_1 - g_\phi(\mathbf{y}_1) - D_1(\mathbf{y}_1 - g_\phi(\mathbf{y}_1))||_2^2. \qquad (26)$$

Same as Eq. 25, with the adoption of a symmetric manner, the consistency loss is represented as:

$$\mathcal{L}_{cons}(\phi) = \frac{1}{2}\big(||\mathbf{y}_1 - g_\phi(\mathbf{y}_1) - D_1(\mathbf{y}_1 - g_\phi(\mathbf{y}_1))||_2^2 + ||\mathbf{y}_2 - g_\phi(\mathbf{y}_2) - D_2(\mathbf{y}_2 - g_\phi(\mathbf{y}_2))||_2^2\big). \quad (27)$$

The noise removal loss function $\mathcal{L}_{noise}$ in Zeroshot-N2N becomes the sum of Eq. 25 and Eq. 27, expressed as:

$$\mathcal{L}_{noise} = \mathcal{L}_{res} + \mathcal{L}_{cons}. \tag{28}$$

### A.5.3 MORE STUDIES ON THE EFFECTIVENESS OF DENOISER

Since NODEs rely on simulation-based training, as the denoiser (Noise Removal module) becomes more complex, it requires more time for training. To mitigate this, we utilize a lightweight 3-layer network (0.085MB) as the denoiser in CLODE. Although the Noise Removal module has fewer parameters, it performs a critical task. It learns to denoise the image at each step (Eq. 16), integrating with the image enhancement process and assisting in predicting fine-grained curve parameter maps at every step.

We train the denoiser concurrently with the image enhancement process to maximize the module's effectiveness. To demonstrate this, we compared three different scenarios: Pre-denoising, CLODE, and Post-denoising. For clarity, 'Pre-denoising' refers to training the denoiser only on the input image, while 'Post-denoising' involves training the denoiser solely on the enhanced image. CLODE outperforms the other approaches, suggesting that it is the optimal scenario among the three (Table 9).

Table 9: Results of denoising ablations.

| Method | PSNR↑ | SSIM↑ |
|---|---|---|
| Pre - denoising | 19.38 | 0.661 |
| CLODE | **19.61** | **0.718** |
| Post - denoising | 19.16 | 0.659 |

The reason for these results is that low-light images have low pixel values, which provide insufficient information for effective denoising. After enhancement, the original noise becomes entangled with the image content. Therefore, we believe that continuous denoising is crucial for effective low-light correction, as noise tends to be amplified with successive exposure enhancements.

**Using existing methods** We observe performance improvements by utilizing existing denoisers to obtain $\tilde{I}_T$. For our experiments, we employed DnCNN (Zhang et al., 2017) and Restormer (Zamir et al., 2022) as denoisers. Quantitatively, as shown in Table 10, Restormer provided a 0.66 gain in SSIM, while DnCNN achieved a 0.1dB gain in PSNR. We also obtained visually superior results, as illustrated in Fig. 11.

Table 10: Quantitative Results of existing denoising methods.

| Method | #params (M) | PSNR | SSIM | LPIPS |
|---|---|---|---|---|
| CLODE | 0.2167 | 19.61 | 0.718 | 0.263 |
| CLODE + DnCNN | 0.8629 | 19.71 | 0.774 | 0.199 |
| CLODE + Restormer | 26.306 | 19.69 | 0.784 | 0.228 |

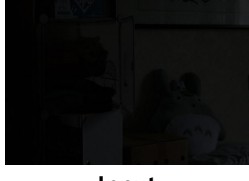 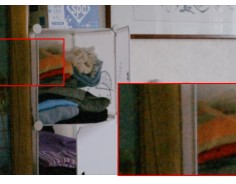 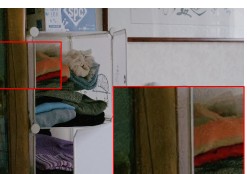 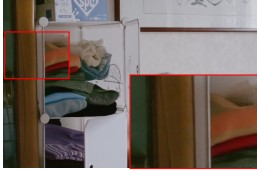

**Input**      **CLODE**      **CLODE + DnCNN**      **CLODE + Restormer**

Figure 11: Visual results with conventional denoising methods.

### A.5.4 MORE QUANTITATIVE RESULTS

We present the comparison results for non-reference metrics, which we did not include in Table 1. Table 11 demonstrates that CLODE outperforms other unsupervised methods in terms of perceptual

quality. Notably, it demonstrates competitive results in terms of BRISQUE and PI, even when compared to state-of-the-art supervised methods. Additionally, in Table 12, we provide LPIPS (Learned Perceptual Image Patch Similarity) (Zhang et al., 2018) performance results. CLODE exhibits superior average LPIPS performance compared to other methods. Moreover, performances on non reference metric for unpaired low-light datasets (DICM (Lee et al., 2013), MEF (Ma et al., 2015), LIME (Guo et al., 2016), NPE (Wang et al., 2013), and VV (Vonikakis et al., 2018)) are provided in Table 13.

Table 11: Comparison results on LSRW (Hai et al., 2023) and LOL (Chen Wei, 2018) in terms of NIQE (Mittal et al., 2012b), BRISQUE (Mittal et al., 2012a), PI (Blau et al., 2018) and Entropy (Chen et al., 2019). Within the unsupervised approaches, the best score is displayed in **Red**. CLODE performs better than all other methods, including supervised methods, in terms of PI (Perceptual Index).

| Training | Method | LSRW | | | | LOL | | | |
|---|---|---|---|---|---|---|---|---|---|
| | | NIQE↓ | BRISQUE↓ | PI↓ | Entropy↑ | NIQE↓ | BRISQUE↓ | PI↓ | Entropy↑ |
| Supervised | Afifi et al. (Afifi et al., 2021b) | 6.655 | 46.645 | 6.470 | 7.065 | 4.966 | 33.546 | 5.741 | 7.173 |
| | RetinexNet (Chen Wei, 2018) | - | - | - | - | 8.871 | 51.813 | 4.955 | 6.835 |
| | URetinexNet (Wu et al., 2022a) | 4.154 | 23.614 | 3.495 | 6.762 | 4.250 | 15.633 | 3.372 | 6.926 |
| | LLFlow (Wang et al., 2022b) | 3.756 | 26.671 | 3.176 | 7.369 | 5.709 | 35.022 | 4.530 | 7.141 |
| | RetinexFormer (Cai et al., 2023) | 3.549 | 15.951 | 3.208 | 7.131 | 3.478 | 17.101 | 3.102 | 7.074 |
| Unsupervised | SCI-easy (Ma et al., 2022) | 3.847 | 25.859 | 3.259 | 6.388 | 7.153 | 12.424 | 5.437 | 5.825 |
| | SCI-medium (Ma et al., 2022) | 3.917 | 22.416 | 3.159 | 6.494 | 7.861 | 25.870 | 4.583 | 6.842 |
| | SCI-difficult (Ma et al., 2022) | 4.368 | 20.692 | 3.851 | 5.975 | 8.060 | 26.823 | 4.664 | 6.675 |
| | RUAS (Liu et al., 2021) | 5.426 | 38.854 | 4.939 | 6.442 | 6.303 | 11.977 | 4.571 | 7.194 |
| | ZeroDCE (Guo et al., 2020) | 3.776 | 23.867 | 3.156 | 6.526 | 7.777 | 27.301 | 4.459 | 6.608 |
| | Night-Enhancement (Jin et al., 2022) | 7.208 | 51.356 | 6.801 | 6.544 | 4.491 | 27.122 | 4.436 | **7.139** |
| | PairLIE (Fu et al., 2023) | **3.684** | 29.816 | 3.426 | 6.923 | **4.083** | 20.592 | 3.052 | 6.823 |
| | **CLODE** | 3.827 | **18.426** | **3.115** | **7.025** | 4.516 | **8.220** | **2.914** | 7.053 |

Table 12: Quantitative results in terms of LPIPS (Zhang et al., 2018). The best average result displays in **Red**.

| Dataset | URetinexNet | RetinexFormer | SCI | RUAS | ZeroDCE | NightEnhancement | PairLIE | **CLODE** |
|---|---|---|---|---|---|---|---|---|
| LSRW | 0.308 | 0.315 | 0.398 | 0.469 | 0.317 | 0.583 | 0.342 | 0.331 |
| LOL | 0.121 | 0.131 | 0.358 | 0.270 | 0.335 | 0.241 | 0.248 | 0.263 |
| SICE | 0.264 | 0.263 | 0.486 | 0.608 | 0.239 | 0.360 | 0.305 | 0.235 |
| MSEC | 0.393 | 0.362 | 0.396 | 0.668 | 0.329 | 0.462 | 0.431 | 0.223 |
| Average | 0.272 | 0.268 | 0.410 | 0.504 | 0.305 | 0.412 | 0.332 | **0.263** |

Table 13: Comparison results on DICM, MEF, LIME, NPE, and VV in terms of NIQE (Mittal et al., 2012b), BRISQUE (Mittal et al., 2012a), NIMA (Talebi & Milanfar, 2018) and Entropy (Chen et al., 2019).

| Method | DICM | | | | MEF | | | | LIME | | | | NPE | | | | VV | | | |
|---|---|---|---|---|---|---|---|---|---|---|---|---|---|---|---|---|---|---|---|---|
| | NIQE↓ | BRISQUE↓ | NIMA↑ | Entropy↑ | NIQE↓ | BRISQUE↓ | NIMA↑ | Entropy↑ | NIQE↓ | BRISQUE↓ | NIMA↑ | Entropy↑ | NIQE↓ | BRISQUE↓ | NIMA↑ | Entropy↑ | NIQE↓ | BRISQUE↓ | NIMA↑ | Entropy↑ |
| SCI-easy | 3.902 | 19.83 | 4.557 | 6.709 | 3.655 | 13.42 | 5.150 | 6.716 | 4.118 | 16.39 | 4.737 | 6.748 | 4.022 | 14.23 | 4.622 | 7.221 | 2.903 | 18.93 | 3.990 | 6.943 |
| SCI-medium | 4.129 | 20.97 | 4.430 | 6.059 | 3.619 | 14.63 | 5.016 | 6.934 | 4.211 | 20.27 | 4.566 | 7.170 | 4.321 | 27.42 | 4.367 | 6.547 | 2.818 | 21.02 | 3.900 | 6.601 |
| SCI-difficult | 4.031 | 20.26 | 4.415 | 6.794 | 3.664 | 13.81 | 4.959 | 7.098 | 4.097 | 16.82 | 4.446 | 7.162 | 4.166 | 16.49 | 4.141 | 7.317 | 2.903 | 18.64 | 3.990 | 7.212 |
| RUAS | 7.153 | 47.01 | 4.178 | 4.413 | 5.408 | 34.59 | 4.732 | 6.248 | 5.420 | 29.58 | 4.391 | 6.896 | 7.063 | 49.83 | 4.171 | 4.559 | 5.230 | 51.07 | 4.015 | 5.191 |
| ZeroDCE | 3.741 | 22.79 | 4.563 | 6.899 | 3.310 | 16.22 | 5.190 | 7.002 | 3.786 | 17.73 | 4.597 | 6.967 | 3.946 | 15.53 | 4.698 | 7.427 | 2.585 | 20.98 | 3.849 | 7.285 |
| ZeroIG | 3.980 | 26.81 | 4.559 | 6.376 | 3.764 | 16.47 | 4.988 | 6.965 | 4.441 | 19.99 | 4.529 | 7.155 | 4.662 | 28.37 | 4.351 | 7.060 | 2.779 | 23.07 | 4.124 | 6.945 |
| PairLIE | 4.297 | 29.37 | 4.360 | 6.925 | 4.203 | 29.81 | 4.702 | 7.183 | 4.547 | 25.26 | 4.389 | 7.082 | 4.238 | 25.90 | 4.509 | 7.286 | 3.295 | 33.82 | 4.100 | 7.324 |
| CLODE | 3.628 | 20.30 | 4.778 | 7.116 | 3.286 | 12.78 | 5.287 | 7.383 | 3.650 | 16.26 | 4.510 | 7.333 | 3.888 | 13.72 | 4.738 | 7.488 | 2.770 | 18.51 | 4.133 | 7.296 |

### A.5.5 COMPARISON WITH OTHER ITERATIVE METHODS

Fig. 12 shows the changes in performance over steps of each curve-adjustment-based method. Each comparison method is retrained for 10 steps in the official code provided by the author. To fix the number of steps in CLODE to 10, we replace CLODE's ODE solver with the Euler method, and referred to it as CLODE-*Euler*. The results show that even within the same number of steps, CLODE-*Euler* performs better than other curve adjustment-based methods. Furthermore, the proposed version, CLODE, demonstrates higher performance compared to other methods in most iterative steps. In case of ReLLIE (Zhang et al., 2021b), it exhibits a decline in performance after 7 steps, emphasizing the need for careful selection of the number of iterative steps itself to achieve optimal result, this makes the method impractical to use.

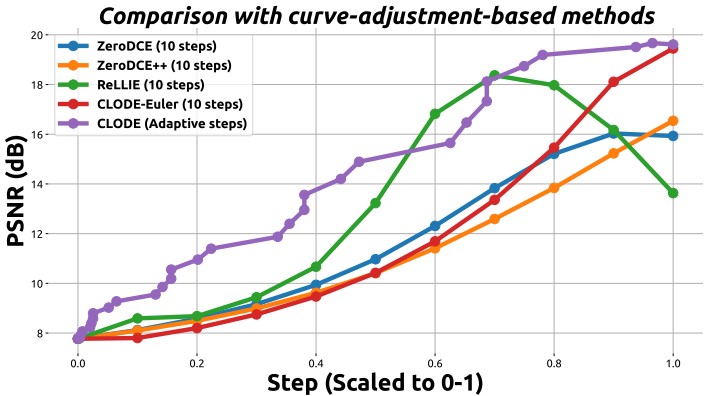

Figure 12: Changes in PSNR (Peak Signal-to-Noise Ratio) over steps of CLODE, CLODE-*Euler*, ReLLIE (Zhang et al., 2021b), ZeroDCE++ (Li et al., 2021), and ZeroDCE (Guo et al., 2020). As CLODE employs a continuous adaptive step according to the input image, we represent the steps by scaling them from 0 to 1. CLODE demonstrates superior performance compared to other methods at almost every step.

## A.6  More visual results

We show additional results for CLODE enhancement that we did not show in the main manuscript due to lack of space. We present additional visual comparison results for PairLIE (Fu et al., 2023) and Night-Enhancement (Jin et al., 2022), which demonstrated the best quantitative performance among the unsupervised methods in Table 1 of the main manuscript, except for our proposed method (CLODE), in Fig. 13. CLODE shows the most robust enhancement results across various image exposure conditions.

Fig. 14, Fig. 15, Fig. 16 and Fig. 17 show the results for CLODE and CLODE† on LOL (Chen Wei, 2018) and SICE (Cai et al., 2018) validation dataset. Additionally, Fig. 18 shows the visual results with different exposures for photos extracted from MSEC (Afifi et al., 2021b) and the internet (Filckr: CC BY-NC 2.0).

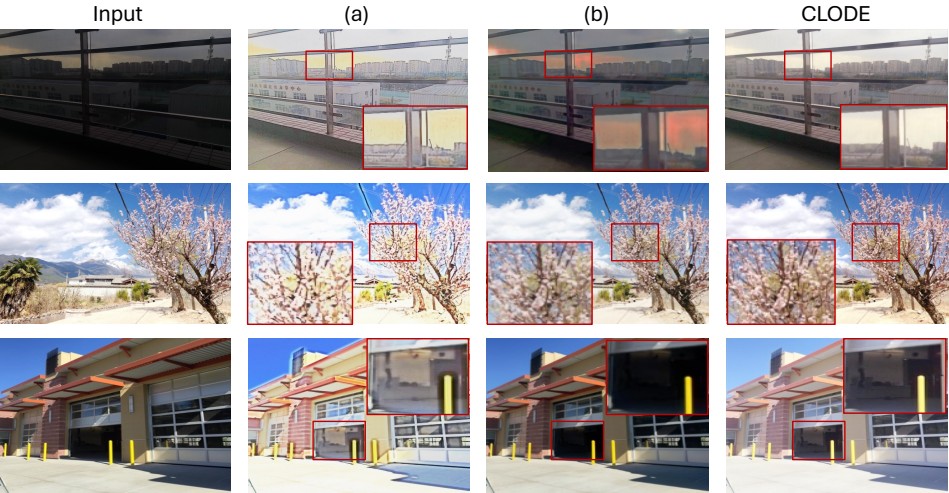

Figure 13: Comparative visualization results with (a) PairLIE (Fu et al., 2023) and (b) night-enhancement (Jin et al., 2022) on LOL (Chen Wei, 2018) and SICE (Cai et al., 2018). Images are taken from LSRW (Hai et al., 2023) and SICE (Cai et al., 2018) Part2.

| Input | CLODE | CLODE† | Ground-Truth |
|---|---|---|---|

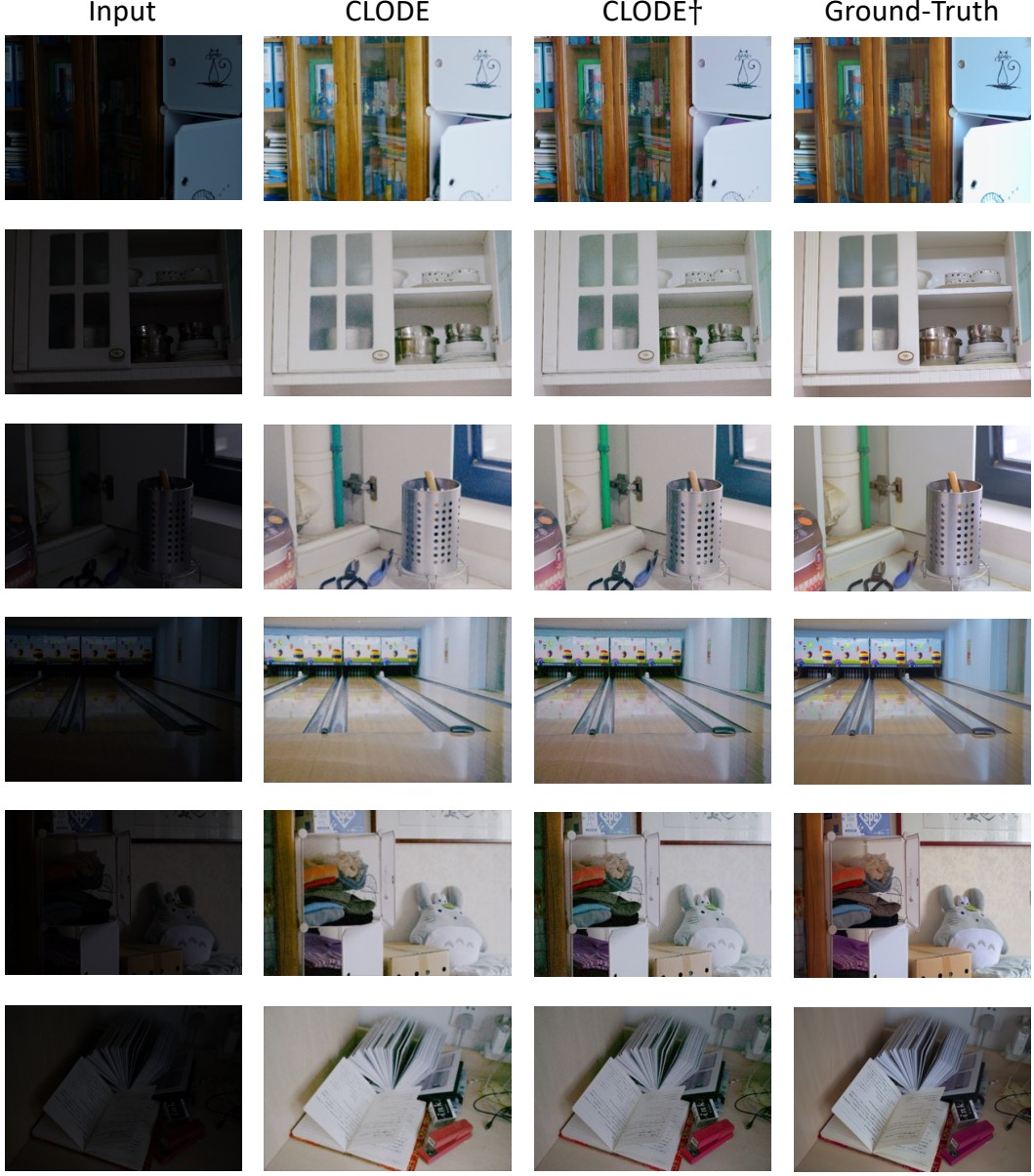

Figure 14: Visualization results on LOL (Chen Wei, 2018). While CLODE demonstrates superior enhancement results, user control with CLODE† produces images that more closely resemble the ground-truth image.

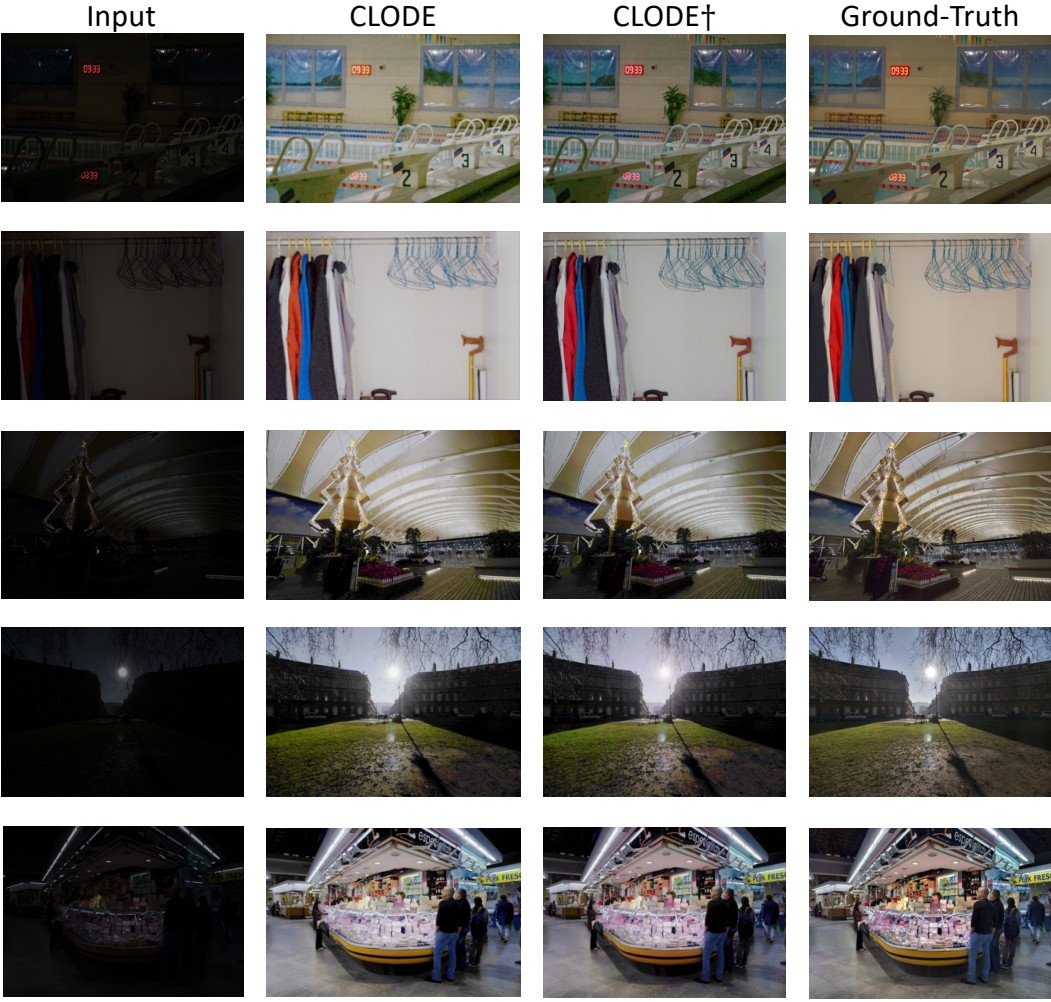

Figure 15: Visualization results on LOL (Chen Wei, 2018) and SICE (Cai et al., 2018). While CLODE demonstrates superior enhancement results, user control with CLODE† produces images that more closely resemble the ground-truth image.

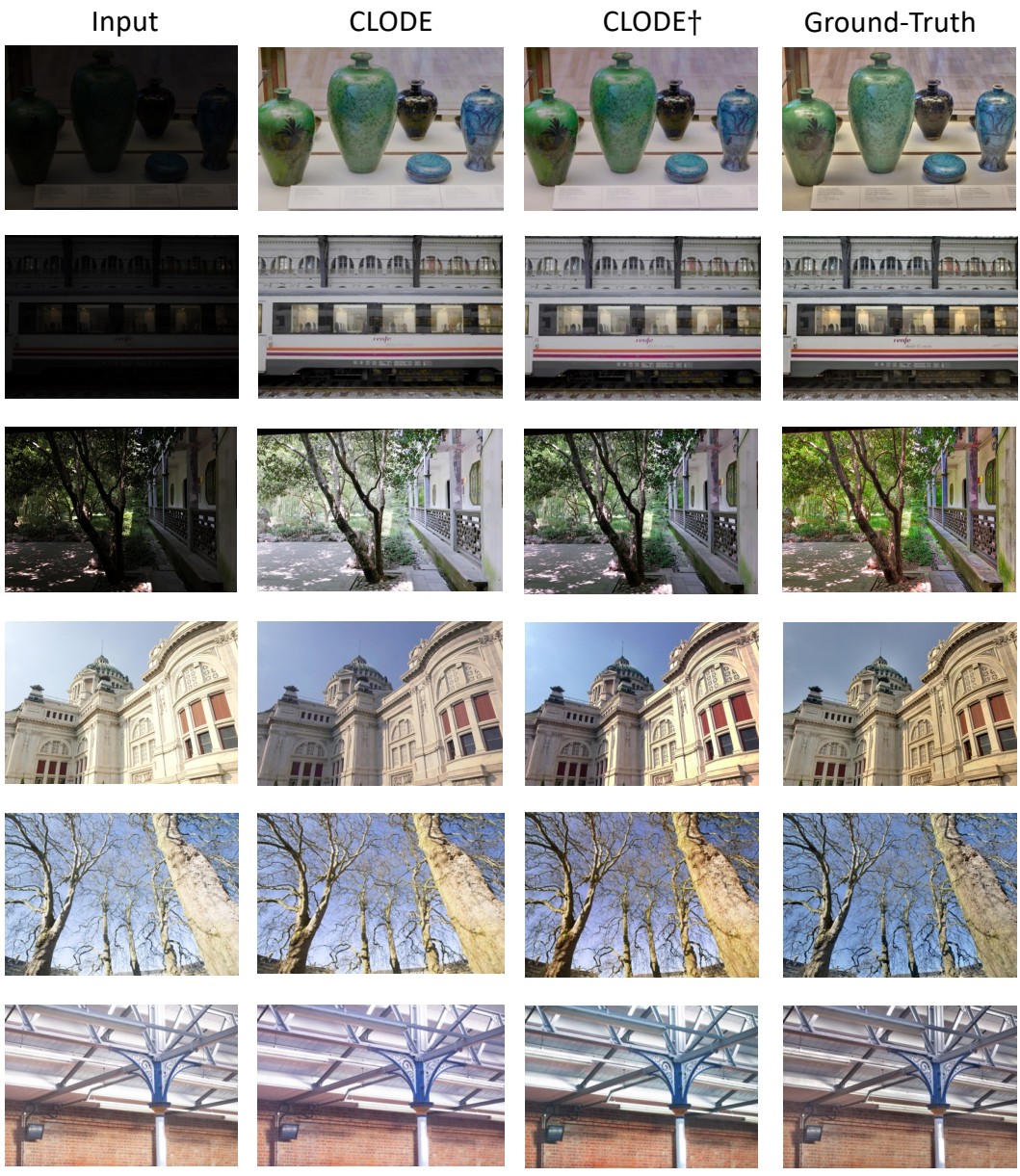

Figure 16: Visualization results on SICE (Cai et al., 2018). While CLODE demonstrates superior enhancement results, user control with CLODE† produces images that more closely resemble the ground-truth image.

| Input | CLODE | CLODE† | Ground-Truth |
|:-:|:-:|:-:|:-:|

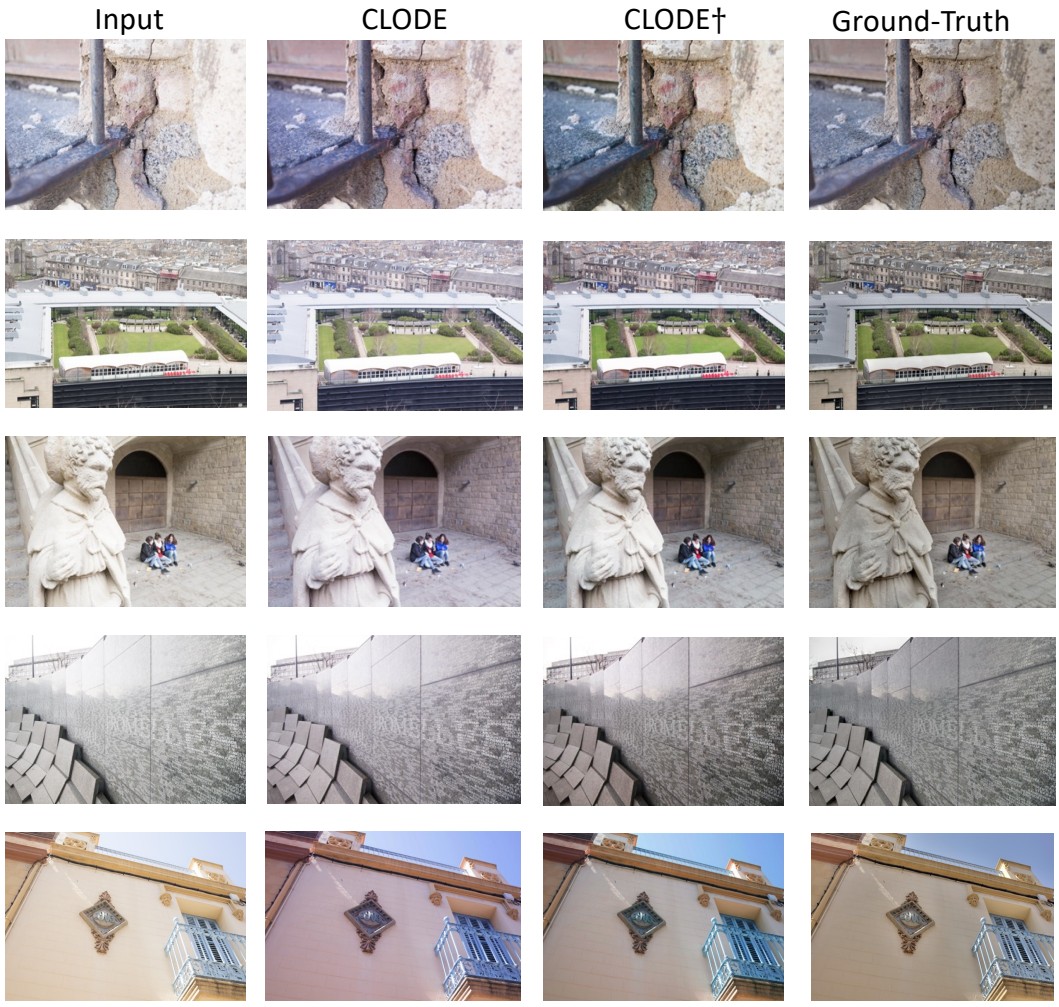

Figure 17: Visualization results on SICE (Cai et al., 2018). While CLODE demonstrates superior enhancement results, user control with CLODE† produces images that more closely resemble the ground-truth image.

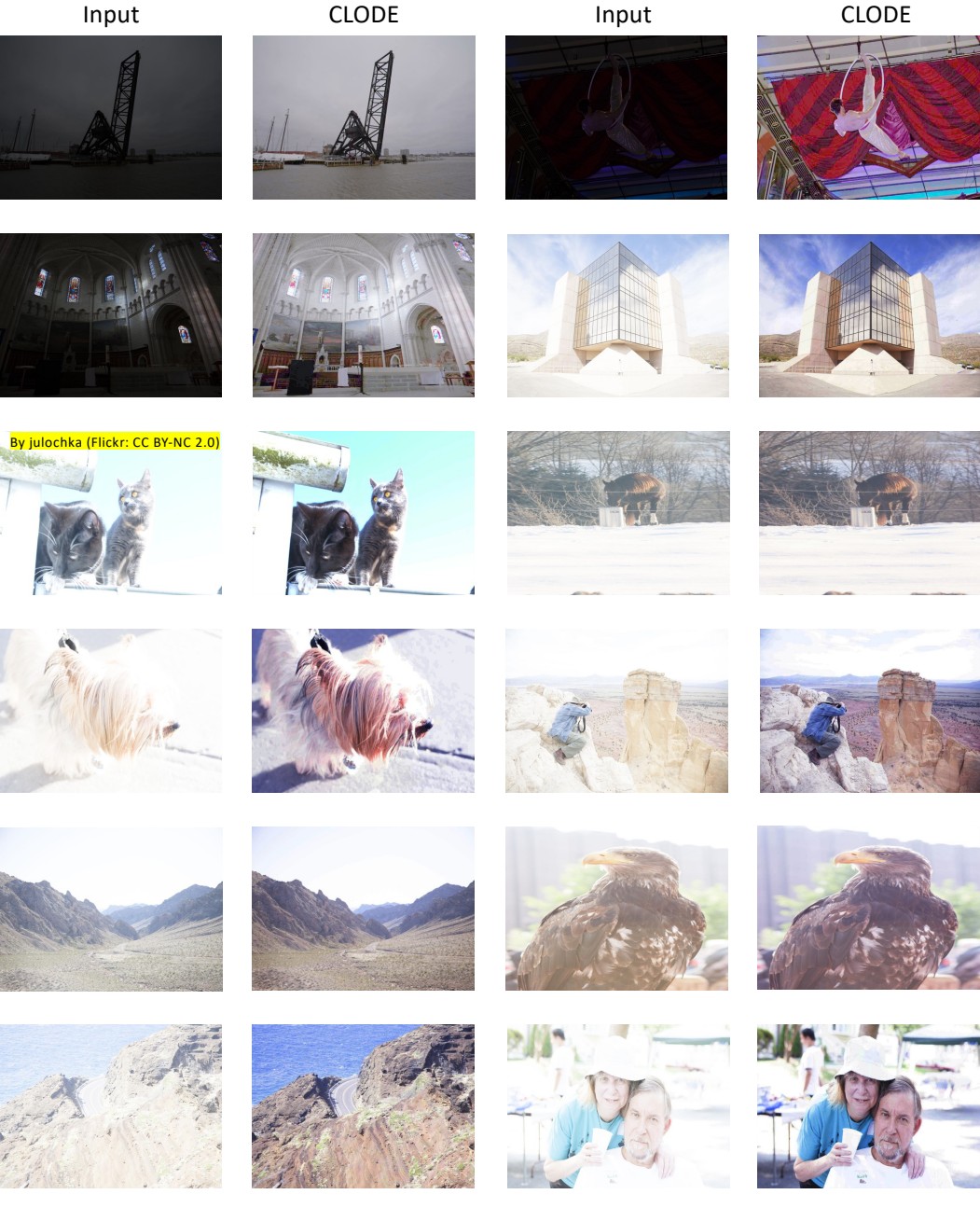

Figure 18: Visualization results on MSEC (Afifi et al., 2021b) and extracted from internet (Flickr by julochka). Even with diverse inputs of various exposures, CLODE show robust result in an unsupervised manner.

