# OpenReview forum: "Continuous Exposure Learning for Low-light Image Enhancement using Neural ODEs"
_ICLR.cc/2025/Conference — ICLR 2025 Spotlight_

### Official Review · Reviewer_NMzG · 2024-11-02

**Soundness:** 3
**Presentation:** 3
**Contribution:** 3
**Rating:** 8
**Confidence:** 5

**Summary:**

This manuscript introduces a CLODE method that enhances low-light images by framing exposure adjustment as a Neural Ordinary Differential Equation (NODE) problem.  It improves on traditional curve-adjustment techniques by automatically adjusting exposure without manual input and converges to an optimal exposure using adaptive ODE solvers. CLODE also includes noise removal and curve parameter estimation, and users can control exposure levels by adjusting integration intervals.
===================================================================

The authors have raised many of my concerns during the rebuttal phase. I am now ready to increase the rating.

**Strengths:**

Strengths of the CLODE method:

1. CLODE’s use of Neural ODEs allows it to dynamically adjust exposure levels based on image-specific needs.
2. The approach is really interesting. Mainly the noise removal block is a very good idea for this type of tasks.
3. Ablation study is really exhaustive.
4. Results are evaluated well.
5. Manuscript is well written and organized. Very easy to follow.
6. Technical content is really well written with proper citations.

Good Paper!!!

**Weaknesses:**

Weaknesses:

1. Novelty is somewhat limited.
2. Comparisons with recent-most SOTA methods should be included.
3. Flowcharts should be improved like Figure 2.
4. Conclusion should be improved although I can understand there is a space limitation. Please make other parts smaller and add some key findings in conclusions.
5. Some more recent references from 2024 is required.

By framing the enhancement process as a Neural ODE, CLODE relies on numerical approximation methods (e.g., dopri5) that inherently carry approximation errors. This can provide solvers with adaptive step sizes but they cannot guarantee analytical precision or stability across all image conditions, particularly in complex cases with high dynamic ranges. This approximation could lead to unpredictable or suboptimal results, especially when the integration steps or the neural network's learned parameters are insufficiently fine-tuned for certain low-light conditions. Can the authors provide some low-light images from real-world dark conditions and demonstrate the method.

**Questions:**

1. Adaptive ODE solvers increase computation time and memory usage. Any idea regarding the number of pixel operations taking place here?

2. The curve parameter estimation module might not generalize well to diverse lighting conditions. Can you provide some generalization test regarding this.

3. Manual adjustment of exposure intervals introduces subjectivity. An ablation can give us more insights.

4. NODE-based adjustments may struggle with high-dynamic-range images, risking instability and suboptimal exposure adjustments. Can you provide the separate loss curves for better understanding of the training dynamics.
==========================================================================

---

> ### Author Response · Authors · 2024-11-18
> **Response to Reviewer NMzG. (PART 1)**
>
> We are glad to hear that you found the paper is well written and organized. We have diligently examined your comments and concerns as a reviewer, and have prepared responses addressing the raised concerns.
>
> **W1. Limited Novelty.**
>
> - We understand the reviewer’s concern that the novelty of our method is limited. **Our method is the first to utilize NODE for low-light image enhancement.** By employing NODE reformulation, we address the limitations of traditional curve adjustment techniques—such as fixed number of iteration and tendency to produce suboptimal enhancement results—in a straightforward yet effective manner.
> - We cautiously wish to assert the novelty of our approach in three aspects: **NODE reformulation**, **Network design**, and **Advantage of NODE**.
>     - **NODE reformulation:** CLODE addresses the limitations of previous curve-adjustment methods that use discrete updates for gradual image enhancement, by reformulating them into NODE. This reformulation facilitates solving for the optimal solution in continuous space and this also results in optimal training. As reference to **Strength1 from Reviewer b2uw**, our method can be interpreted as a creative application of NODEs, which is a contribution to the low-light image enhancement field.
>     - **Network design:** In addition to reorganizing the iterative cure-adjustment formula, we designed a suitable network consisting of Noise Removal module and Curve Parameter Estimation module as shown in Sec 3.2.1 ODE function. Each module is sophisticatedly tailored for NODE-specific iterative noise removal and curve map estimation.
>     - **Advantage of NODE:** As noted in Sec. 3.3, *User Controllable Design*, users can manually adjust the integration interval by changing the final state value, allowing them to output images with the preferred exposure level. By leveraging the features of NODE, as illustrated in Figure 3 of the main paper, this user-controllability feature enhances the potential of our model. This capability represents one of the most effective and practical aspects of incorporating NODE, as it facilitates customized brightness adjustments to accommodate individual preferences.
> - In the context of unsupervised methods in low-light image enhancement, we firmly believe that the first attempt of transporting discrete curve-adjustment method problem to continuous space by neural ordinary differential equations, designing adequate network architecture for the NODE method, and providing high visual performance, user-controllability, constitute meaningful contributions. Accordingly, we kindly ask for a reconsideration of the novelty of our work.
>
>
> **W2, W5. Need More Recent References.**
>
> - In the main paper, **Retinexformer** is referred, a transformer-based state-of-the-art method and the second-best performer in the NTIRE 2024 low-light challenge [1*], as a strong supervised comparison method.
> - We are thankful for Reviewer NMzG’s concern and we are going to add additional strong supervised diffusion-based baselines **GSAD [2\*]** and **PyDiff [3\*]** in terms of PSNR. Even when compared to diffusion-based supervised methods, CLODE demonstrates **competitive performance**.
> - Additionally, we included a comparison with **ZeroIG** [4*], an unsupervised learning method introduced at **CVPR 2024**. The results show that the performance of both CLODE and CLODE+ significantly surpasses that of **ZeroIG [4\*]**.
> - We will include performances of  three methods in our final revised version.
> |Type|Model|LSRW/LOL|LSRW/LOL (GT Mean)|
> |-----|-------|:-----------:|:------------------------:|
> |supervised|GSAD [2*]|17.37/23.01|19.51/27.60|
> |supervised|PyDiff [3*]|17.00/20.49|20.11/26.99|
> |**Unsupervised**|ZeroIG [4*]|12.42/19.60|18.22/22.17|
> |**Unsupervised**|**CLODE**|**17.28/19.61**|**20.60/23.16**|
> |**Unsupervised**|**CLODE+**|**20.77/23.58**|**20.94/24.47**|
>
>
> **W3. Flowcharts should be improved like Figure 2.**
>
> - For better understanding of overall process and network details, we will revise the final version to include the network architecture from **Figure 7** as part of the main figure.
> - **Additionally, the improved Figure 2 can be found in the updated Rebuttal version.**
>
> **W4. Conclusion should be improved.**
>
> - Yes, the initial version was somewhat brief due to space limitations. **We revised the conclusion** to emphasize CLODE’s novelty in the reformulation of curve-adjustment and user-controllability, and updated it in the rebuttal version. Thank you for your valuable feedback.

---

> ### Author Response · Authors · 2024-11-18
> **Response to Reviewer NMzG. (PART 2)**
>
> **W6. Limitations in CLODE for Complex Low-Light Conditions**
>
> - To address Reviewer **NMzG**‘s concerns, we evaluated the performance of our method in terms of  **BRISQUE↓/Entropy↑** for the DICM[5*], MEF[6*], LIME[7*], NPE[8*], and VV[9*] datasets, which are commonly utilized for assessing complex low-light scenarios.
> - **Our method, CLODE demonstrates superior performance on non-reference metrics across all five realistic datasets, except for the BRISQUE value on the DICM dataset (2nd best in DICM).**
> |Model|DICM [5*]|MEF [6*]|LIME [7*]|NPE [8*]|VV [9*]|
> |-------|:---------:|:-------:|:---------:|:-------:|:------:|
> |SCI-easy|**19.83**/6.709|13.42/6.716|16.39/6.748|14.23/7.221|18.93/6.943|
> |SCI-medium|20.97/6.059|14.63/6.934|20.27/7.170|27.42/6.547|21.02/6.601|
> |SCI-difficult|20.26/6.794|13.81/7.098|16.82/7.162|16.49/7.317|18.64/7.212|
> |RUAS|47.01/4.413|34.59/6.248|29.58/6.896|49.83/4.559|51.07/5.191|
> |ZeroDCE|22.79/6.899|16.22/7.002|17.73/6.967|15.53/7.427|20.98/7.285|
> |ZeroIG [4*]|26.81/6.376|16.47/6.965|19.99/7.155|28.37/7.060|23.07/6.945|
> |**CLODE**|20.30/**7.116**|**12.78/7.383**|**16.26/7.333**|**13.72/7.488**|**18.51/7.296**|
> - Additionally, the results for LPIPS and non-reference metrics (NIQE, BRISQUE, PI, Entropy) on the LOL and LSRW datasets can be found in Appendix Tables 11 and 12.
> - **The results for LPIPS and non-reference metrics also demonstrate that CLODE is the most robust.**
>
> **Q1. Computational Cost of using Adaptive Solver.**
>
> - As Reviewer commented, increasing computation time and memory usage due to adaptive solver should be considered.
> - In Sec. 5, *Limitation* and Table 6, we covered this issue that CLODE incurs higher computational costs compared to lightweight unsupervised methods due to the iterative solutions of NODE. However, it achieves significant performance gains over existing unsupervised methods, making it comparable to supervised approaches.
> - To overcome computational cost due to adaptive solver, we present the results of CLODE-S, as introduced in Sec. 5, *Limitation*. CLODE-S is a simplified network with a 1x1 convolutional 2-layer structure, without the noise removal module. The detailed architecture can be found in Appendix A.1.2, Figure 7 (b). **CLODE-S outperforms other methods in all computational cost metrics**. While it shows a performance drop compared to CLODE, it demonstrates robust performance compared to previous unsupervised methods.
> - In the following, we present an extended version of Table 6. **We provide the additional FLOPs results for processing a 3x256x256 image to support additional computational cost measurements for your reference.**
> - Adaptive ODE solvers in our CLODE dynamically adjust the number of integration steps based on the complexity of the input. As mentioned in Appendix Sec. A.4, *CLODE Step Statistics Across Exposure Conditions*, the average forward steps of CLODE are 21.12 under low-light condition, with each step requiring 2.65 GFLOPs. Consequently, processing a single image requires 58.3 GFLOPs (2.65 × 22).
> - Light version of CLODE, CLODE-S applies an Euler solver for speed optimization, using 10 steps. Each step requires **0.026** **GFLOPs**, resulting in a total of **0.26 GFLOPs** per image which is the smallest value among methods in the table.
> - In summary, although CLODE requires higher computational cost due to its high-order solution process, it remains comparable to other methods, making it practical to use, and within CLODE-S we provide a low computational cost version of our method.
> - We present an extended version of Table 6 from the main manuscript:
> |Type|Model|PSNR/SSIM|#Params (M)|Time (S)|FLOPs (G)|
> |-----|--------|--------------|:------------:|---------|----------|
> |Supervised|RetinexNet|15.49/0.355|0.4446|0.337|587.47|
> |Supervised|LLFlow|17.52/0.509|38.859|0.144|286.34|
> |Supervised|RetinexFormer|17.76/0.517|1.6057|0.072|16.71|
> |Unsupervised|Night-Enhancement|14.24/0.472|67.011|0.035|52.58|
> |Unsupervised|PairLIE|16.97/0.498|0.3417|0.008|22.35|
> |Unsupervised|**CLODE**|17.28/0.533|**0.2167**|**0.056**|58.3 (=2.65 x 22)|
> |Unsupervised|**CLODE-S**|16.97/0.457|**0.0004**|**0.005**|**0.26 (=0.026 x 10)**|

---

> > ### Author Response · Authors · 2024-11-18
> > **Response to Reviewer NMzG. (PART 3)**
> >
> > **Q2. Discussion about Generalization capability of Curve Parameter Estimation module**
> >
> > - First, we would like to emphasize that CLODE’s parameter size is only 0.2167M, which is significantly smaller than other supervised methods, thereby reducing concerns about overfitting. Additionally, we address the generalization capability from both quantitative and qualitative aspects.
> > - **Quantitive aspect:**
> >     - To demonstrate the generalization performance, we also evaluate CLODE trained on the LOL dataset on the LSRW dataset, as shown in Table 1 of our main manuscript.
> >     - LSRW is a dataset with different hardware configurations (Nikon, Huawei) and various exposure level compared to the LOL dataset.
> >     - Elaborating on the experiments in Table 1, all methods were trained on the LOL dataset and evaluated on both the LSRW and LOL datasets.
> >     - **CLODE demonstrates robust performance on the LSRW dataset, showcasing superior generalization capabilities compared to other methods even without training on the LSRW dataset,**
> >     - **These results indicate that CLODE not only overcomes the overfitting limitations faced by other methods but also exhibits superior generalization performance in low-light conditions.**
> >     - Additionally, Table 2 presents the performance results on the SICE dataset, which includes a mix of under- and over-exposed conditions.
> >     - We provide training and evaluation results for CLODE and other low-light methods under these complex scenarios, where CLODE demonstrates superior performance compared to other low-light approaches. This highlights CLODE’s ability to handle complex exposure scenarios more effectively than other models.
> >     - For clarity, we have reorganized Table 1 (use GT mean) and included **ZeroIG [4\*]** from CVPR 2024.
> > |Type|Model|LOL(train)→LSRW(eval)|LOL(train)→LOL(eval)|
> > |-----|-------|:-------------------------:|:-----------------------:|
> > |Supervised|LLFlow|18.68/0.518|24.99/0.871|
> > |Supervised|RetinexFormer|19.15/0.529|27.18/0.850|
> > |Unsupervised|ZeroDCE|18.87/0.467|20.99/0.596|
> > |Unsupervised|ZeroIG [4*]|13.30/0.341|22.17/0.771|
> > |Unsupervised|**CLODE**|**20.60/0.557**|**23.16/0.752**|
> > |Unsupervised|**CLODE$\dagger$**|**20.94/0.568**|**24.47/0.759**|
> >
> > - **Qualitative aspect**: To support the generalization capability of our method, we include a range of visual examples in the Appendix. Figure 13 presents the results of our method compared to others on under-exposed, normal, and over-exposed images.
> >
> > **Q3. About Manual Adjustment.**
> >
> > - First, we would like to cautiously mention that CLODE, even without user-control (manual adjustment), is already state-of-the-art in unsupervised manner.
> > - In **Table 1** and **2 of main manuscript**, CLODE+ was determined based on the average values of images selected by two experts in the field.
> > - To address Reviewer **NMzG**’s curiosity, we conducted an extra user study on the LOL dataset with 30 non-expert participants who had no prior knowledge of the ground-truth images. From the optimal state provided by CLODE, five images were generated for the participants by shifting the time steps by -1, -0.5, 0, +0.5, and +1, respectively.
> > |Model|PSNR↑/SSIM↑|BRISQUE↓/PI↓|
> > |-------|:---------------:|:---------------:|
> > |CLODE (automatic)|19.61/0.718|8.220/**2.914**|
> > |User Study (30 non-experts)|21.66/0.738|**8.210**/2.970|
> > |CLODE$\dagger$ (2 experts)|**23.58/0.754**|8.228/3.068|
> > - The results of the user study fall between the values for CLODE and CLODE$\dagger$. **Since CLODE already produces high-quality images through optimal solutions and empirically achieves more appealing images near the optimal state**, finding user-preferred images is challenging. Furthermore, the reason for the lower results in the user study is due to exceptionally dark reference images, even though the results from CLODE$\dagger$ look better. Therefore, there is no significant difference in terms of visual quality (BRISQUE↓, PI↓).

---

> > > ### Author Response · Authors · 2024-11-18
> > > **Response to Reviewer NMzG. (PART 4)**
> > >
> > > **Q4. Loss curves of training dynamics.**
> > >
> > > - Thanks to Reviewer’s comment, we provide the loss curves and metrics (PSNR, SSIM) curves for training dynamics in **Appendix Sec.A.7.** *Training Dynamics: Loss and Metric Curves*. (**Figure 19, 20, 21**), updated in the rebuttal version.
> > > - As noted by Reviewer **NMzG**, NODE-based training is highly sensitive, and using an excessively high learning rate can lead to failure during the initial training phase. Empirically, we set the learning rate to $1 \times 10^{-5}$, and as shown in the Figure 19, 20, 21, both the loss curve and performance metrics remain stable throughout the training process.
> > > - In Figure 19, although the reductions in $\mathcal L_{col}$ and $\mathcal L_{exp}$ are minimal, they serve as constraints and successfully converge during training.
> > > - Details regarding the ablation study on the loss functions can be found in **Appendix Sec. A2,** *Impact of Each Loss Function*, with the results presented in **Figure 8** and **Table 7.**
> > > - Additionally, the inference dynamics under varying exposure conditions can be found in **Appendix Sec. A.4, Figure 10 (Left)**.
> > > - We hope this helps in better understanding our CLODE!
> > >
> > > > **Reference**
> > > >
> > >
> > > [1*] Liu, Xiaoning, et al. "NTIRE 2024 challenge on low light image enhancement: Methods and results." In CVPRW, 2024.
> > >
> > > [2*] Hou, Jinhui, et al. "Global structure-aware diffusion process for low-light image enhancement." In NeurIPS, 2023.
> > >
> > > [3*] Zhou, Dewei, Zongxin Yang, and Yi Yang. "Pyramid diffusion models for low-light image enhancement." In IJCAI, 2023.
> > >
> > > [4*] Shi, Yiqi, et al. "ZERO-IG: Zero-Shot Illumination-Guided Joint Denoising and Adaptive Enhancement for Low-Light Images." In CVPR, 2024.
> > >
> > > [5*] Chulwoo Lee, Chul Lee, and Chang-Su Kim. Contrast enhancement based on layered difference representation of 2d histograms. TIP, 2013.
> > >
> > > [6*] Kede Ma, Kai Zeng, and Zhou Wang. Perceptual quality assessment for multi-exposure image fusion. TIP, 2015.
> > >
> > > [7*] Xiaojie Guo, Yu Li, and Haibin Ling. Lime: Low-light image enhancement via illumination map estimation. TIP, 2016.
> > >
> > > [8*] Shuhang Wang, Jin Zheng, Hai-Miao Hu, and Bo Li. Naturalness preserved enhancement algorithm for non-uniform illumination images. TIP, 2013
> > >
> > > [9*] Vassilios Vonikakis, Rigas Kouskouridas, and Antonios Gasteratos. On the evaluation of illumination compensation algorithms. Multimedia Tools and Applications, 2018.

---

> > > > ### Comment · Reviewer_NMzG · 2024-11-21
> > > >
> > > > Dear Authors,
> > > >
> > > > Thank you very much for addressing my comments. Now I am happy to increase the points.

---

> > > > > ### Author Response · Authors · 2024-11-22
> > > > >
> > > > > To Reviewer **NMzG**,
> > > > >
> > > > > We sincerely appreciate your thoughtful reconsideration and the adjustment of your score. Your recognition of our work’s contributions and potential means a great deal to us. Thank you for your valuable feedback and support.
> > > > >
> > > > > In the final stage, we will work diligently to incorporate suggestions from other reviewers as well and update the final version accordingly. Once again, thank you for your valuable input.
> > > > >
> > > > > Best regards,
> > > > > Authors of #4698

---

### Official Review · Reviewer_Asgg · 2024-11-03

**Soundness:** 3
**Presentation:** 3
**Contribution:** 2
**Rating:** 8
**Confidence:** 4

**Summary:**

In this article, the author discovers that the traditional curve-based iterative methods cannot guarantee the convergence of the enhancement process, leading to sensitivity to the number of iterations and limited performance. The author regards the iterative curve adjustment process as a dynamic system and, for the first time, formulates it as a neural ordinary differential equation (NODE) to learn the continuous dynamics of underlying images. Specifically, this method leverages NODE to harness the continuous dynamics within iterative methods, achieving better convergence than discrete space methods. Finally, the article demonstrates superior performance over current unsupervised low-light image enhancement methods on various benchmark datasets.

**Strengths:**

1. In this article, for the first time, the iterative curve adjustment update process is regarded as a dynamic system and formulated as a neural ordinary differential equation (NODE) to learn the continuous dynamics of underlying images. This approach offers an alternative perspective for addressing low-light issues；
2. The article employs a substantial amount of mathematical language to demonstrate how to transform the iterative curve adjustment update into a continuous process, which enhances the logical strength of the paper；
3. The language of the article is quite fluent and adheres to English writing standards；

**Weaknesses:**

1. Since this article addresses issues in low-light scenarios, presenting more objective results in complex scenarios would enhance the persuasiveness of the article.

**Questions:**

1. Since this article addresses issues in low-light scenarios, presenting more objective results in complex scenarios would enhance the persuasiveness of the article.

---

> ### Author Response · Authors · 2024-11-12
>
> - Thank you for taking the time to review our paper and for providing your feedback.We would like to kindly note that we have included all the benchmark experiments typically conducted in low-light image enhancement.
> - In Table 2, we present results on the complex SICE dataset using various metrics to provide a comprehensive evaluation. Unlike methods that measure performance with only a specific exposure from the SICE dataset, we leverage all available exposures to evaluate full complex scenarios. As a result, our method demonstrates performance on par with supervised learning approaches in these challenging SICE scenarios.
> - However, the current request outlined in your review seems a bit ambiguous.  We would greatly appreciate it if you could provide more specific details regarding additional experiments or analyses you would like to see. This would help us address your concerns effectively during the discussion period. Additionally, **we would be grateful if you could provide clearer reasons for the strong rejection recommendation**, as this would help us better understand your perspective and improve our work.

---

> ### Author Response · Authors · 2024-11-18
> **Response to Reviewer Asgg.**
>
> - First, we would like to revisit the results presented in Table 2 of our main manuscript.
> - Alongside low-light conditions, we also conducted experiments on complex scenarios, as mentioned by Reviewer **Asgg**, using under- and over-exposure data as part of the training set for Table 2.
> - The SICE dataset includes 229 images with diverse lighting conditions ranging from under- to over-exposure. To evaluate performance under complex scenarios, we utilized all lighting conditions in the SICE dataset as benchmarks.
> - As shown in **Table 2 of the main manuscript**, the results on the multi-exposure dataset demonstrate that CLODE performs significantly more robustly than other methods, **proving its effectiveness in handling complex scenarios**.
> - Additionally, we provide results on **five real-world datasets.** in term of **BRISQUE↓ / Entropy↑**.
> - **Our method, CLODE demonstrates superior performance on non-reference metrics across all five realistic datasets, except for the BRISQUE value on the DICM dataset (2nd best in DICM).**
> |Model|DICM [1*]|MEF [2*]|LIME [3*]|NPE [4*]|VV [5*]|
> |-------|-----------|----------|----------|---------|--------|
> |SCI-easy|**19.83**/6.709|13.42/6.716|16.39/6.748|14.23/7.221|18.93/6.943|
> |SCI-medium|20.97/6.059|14.63/6.934|20.27/7.170|27.42/6.547|21.02/6.601|
> |SCI-difficult|20.26/6.794|13.81/7.098|16.82/7.162|16.49/7.317|18.64/7.212|
> |RUAS|47.01/4.413|34.59/6.248|29.58/6.896|49.83/4.559|51.07/5.191|
> |ZeroDCE	|22.79/6.899|16.22/7.002|17.73/6.967|15.53/7.427|20.98/7.285|
> |ZeroIG [6*]|26.81/6.376|16.47/6.965|19.99/7.155|28.37/7.060|23.07/6.945|
> |**CLODE**|20.30/**7.116**|**12.78/7.383**|**16.26/7.333**|**13.72/7.488**|**18.51/7.296**|
>
> - We also present the results for **NIQE$\downarrow$** and **NIMA$\uparrow$** as shown below. Excluding the NIMA score on the LIME dataset, **CLODE demonstrates the best performance.**
> |Model|DICM [1*] |MEF [2*]|LIME [3*]|NPE [4*]|VV [5*]|
> |-------|----------|---------|------------|----------|---------|
> |SCI-easy|	3.902/4.557|	3.655/5.150|	4.118/4.737|	4.022/4.622|	2.903/3.990|
> |SCI-medium|	4.129/4.430|	3.619/5.016|	4.211/4.566|	4.321/4.367|	2.818/3.900|
> |SCI-difficult|	4.031/4.415|	3.664/4.959|	4.097/4.446|	4.166/4.141|	2.903/3.990|
> |RUAS|	7.153/4.178|	5.408/4.732|	5.420/4.391|	7.063/4.171|	5.230/4.015|
> |ZeroDCE|	3.741/4.563|	3.310/5.190|	3.786/**4.597**|	3.946/4.698|	2.585/3.849|
> |ZeroIG [6*]|	3.980/4.559|	3.764/4.988|	4.441/4.529|	4.662/4.351|	2.779/4.124|
> |CLODE|	**3.628/4.778**|	**3.286/5.287**|	**3.650**/4.510|	**3.888/4.738**|	**2.770/4.133**|
>
> - The visual results for complex scenarios (e.g., over-exposed conditions) **can also be found in Appendix, Figures 13 and 18**.
> - Additionally, we kindly ask you to refer to the questions and responses provided for reviewers **b2uw** and **NMzG**.
> - In the reviewer's comments, we were unable to identify sufficient information to fully address the rebuttal. Due to this limitation, designing specific experiments was challenging. However, we conducted as many relevant experiments as possible to address the reviewer’s concerns. **We kindly ask you to review our rebuttal letter and consider revising our score if our responses have clarified the points of concern in our paper**.
>
> > **References**
> >
>
> [1*] Chulwoo Lee, Chul Lee, and Chang-Su Kim. Contrast enhancement based on layered difference representation of 2d histograms. TIP, 2013.
>
> [2*] Kede Ma, Kai Zeng, and Zhou Wang. Perceptual quality assessment for multi-exposure image fusion. TIP, 2015.
>
> [3*] Xiaojie Guo, Yu Li, and Haibin Ling. Lime: Low-light image enhancement via illumination map estimation. TIP, 2016.
>
> [4*] Shuhang Wang, Jin Zheng, Hai-Miao Hu, and Bo Li. Naturalness preserved enhancement algorithm for non-uniform illumination images. TIP, 2013
>
> [5*] Vassilios Vonikakis, Rigas Kouskouridas, and Antonios Gasteratos. On the evaluation of illumination compensation algorithms. Multimedia Tools and Applications, 2018.
>
> [6*] Shi, Yiqi, et al. "ZERO-IG: Zero-Shot Illumination-Guided Joint Denoising and Adaptive Enhancement for Low-Light Images." In CVPR, 2024.

---

> > ### Comment · Reviewer_Asgg · 2024-11-27
> >
> > Dear Authors, thank you very much for addressing my comments. Happy to increase the points.

---

> > > ### Author Response · Authors · 2024-11-27
> > >
> > > To Reviewer **Asgg**,
> > >
> > > We appreciate to address your comments and the adjustment of your score.
> > >
> > > We will carefully incorporate suggestions to further refine the final version.
> > >
> > > Best regards,
> > >
> > > Authors of #4698

---

### Official Review · Reviewer_b2uw · 2024-11-03

**Soundness:** 3
**Presentation:** 2
**Contribution:** 3
**Rating:** 6
**Confidence:** 4

**Summary:**

This paper introduces a novel approach for enhancing low-light images by formulating the problem as a Neural Ordinary Differential Equations (NODE) problem. The authors propose CLODE, a dynamic system that leverages continuous dynamics of latent images to improve exposure levels in images. The method is unsupervised, which is particularly useful given the difficulty of obtaining paired low-light and well-exposed images for supervised learning.

**Strengths:**

1. The paper presents a creative application of NODEs to the problem of low-light image enhancement, which is a contribution to the field.

2. Addressing the challenge of lacking paired datasets, the unsupervised nature of CLODE makes it highly applicable to real-world scenarios where ground truth data is scarce.

3. The paper provides a thorough explanation of the CLODE model, including its theoretical foundations and implementation details.

4. The method achieves competitive results across multiple benchmarks, which is a strong point of its potential impact.

**Weaknesses:**

1. The paper does not explicitly address the computational cost of solving NODEs compared to other methods, which is an important consideration for practical applications. Authors should report this results compared with other similar methods.

2. The paper does not discuss the potential for overfitting, especially since the model is tailored to low-light images, which may have unique characteristics.

3. In my opinion, the author's comparative experiment is still not sufficient. It should be compared on more real benchmarks to highlight the superiority of the method. The current comparison seems to be insufficient, including the following:

[1] Contrast enhancement based on layered difference representation.

[2] Perceptual quality assessment for multi-exposure image fusion.

[3] Structure-revealing low-light image enhancement via robust retinex model.

[4] Naturalness preserved enhancement algorithm for non-uniform illumination images.

[5] On the evaluation of illumination compensation algorithms.

4. There are many different paradigms for unsupervised methods. Compared with this type of method, is the author's method more scalable? If it can be improved on other methods, I think this article will have higher value.

In general, I will make further adjustments based on the author's rebuttal.

**Questions:**

Please refer to weaknesses.

---

> ### Author Response · Authors · 2024-11-18
> **Response to Reviewer b2uw. (PART 1)**
>
> Thank you very much for thoroughly reviewing our paper. We appreciate your feedback. To address the concerns you raised, we are providing several experimental results and our perspectives.
>
> **W1. Additional Computational Cost.**
> - For the computational cost, Table 6 in the main paper presents the number of parameters and execution time for our method compared to others. This is further discussed in Sec. 5 (Limitations) of the main manuscript.
> - **To further address reviewer’s concern, we provide the FLOPs results for processing a 3x256x256 image to support additional computational cost measurements.**
> - Specifically, the adaptive ODE solvers in our CLODE dynamically adjust the number of integration steps based on the complexity of the input. As mentioned in Appendix Sec. A.4, *CLODE Step Statistics Across Exposure Conditions*, the  average forward steps of CLODE are 21.20 under low-light condition, with each step requiring 2.65 GFLOPs. Consequently, we calculated that 58.3 GFLOPs (2.65 × 22) are required to process a single image.
> - Despite incorporating higher-order solution processes, **CLODE operates with lower computational cost compared to other methods, making it practical for use.**
> - In addition, we present the results of **CLODE-S** which is a smaller version of CLODE, as introduced in Sec. 5, *Limitation,* **that outperforms other methods in all computational cost metrics**. While this smaller version shows a performance drop compared to CLODE, it demonstrates robust performance compared to previous unsupervised methods.
> - CLODE-S applies an Euler solver for speed optimization, using 10 fixed steps. Thus, each step requires **0.026** **GFLOPs**, resulting in a total of **0.26 GFLOPs** per image.
> - In detail, CLODE-S is a simplified network with a 1x1 convolutional 2-layer structure, without the noise removal module. The detailed architecture can be found in Appendix A.1.2, Figure 7 (b).
> - We present an extended version of Table 6 from the main manuscript:
> |Type|Model|PSNR/SSIM|#Params (M)|Time (S)|FLOPs (G)|
> |-----|-------|---------------|:-------------:|---------|-----------|
> |Supervised|RetinexNet|15.49/0.355|0.4446|0.337|587.47
> |Supervised|LLFlow|17.52/0.509|38.859|0.144|286.34|
> |Supervised|RetinexFormer|17.76/0.517|1.6057|0.072|16.71|
> |Unsupervised|Night-Enhancement|14.24/0.472|67.011|0.035|52.58|
> |Unsupervised|PairLIE|16.97/0.498|0.3417|0.008|22.35|
> |Unsupervised|**CLODE**|17.28/0.533|**0.2167**|**0.056**|58.3 (=2.65 x 22)|
> |Unsupervised|**CLODE-S**|16.97/0.457|**0.0004**|**0.005**|**0.26 (=0.026 x 10)**|
>
> **W2. Overfitting Problem (Discussion about Generalization)**
> - As noted by the Reviewer **b2uw**, low-light enhancement methods often face overfitting issues due to the limited train dataset.
> - CLODE’s parameter size is only 0.2167M, which is significantly smaller than other supervised methods with larger parameter sizes, reducing concerns about overfitting.
> - The main experiment in Table 1, demonstrates capability of our method on low-light datasets.
> - In Table1  of our main manuscript, we evaluate CLODE trained on the LOL dataset and test it on both the LOL and LSRW datasets to demonstrate its generalization performance under low-light conditions.
> - Note that LSRW is a dataset with different hardware configurations (Nikon, Huawei) and various exposure level compared to the LOL dataset. Elaborating on the experiments in Table 1, all compared methods were also trained on the LOL dataset and evaluated on both the LSRW and LOL datasets.
> - **CLODE demonstrates robust performance on the LSRW dataset, showcasing superior generalization capabilities compared to other methods even without training on the LSRW dataset.**
> - Moreover, the promising performance also demonstrated on the LOL dataset.
> - **These results suggest that CLODE  not only overcomes the overfitting limitations faced by other methods but also exhibits superior generalization performance in low-light conditions.**
> - For clarity, we have reorganized Table 1 (use GT mean) and included a recent reference **ZeroIG [7\*]** in CVPR 2024:
> |Type|Model|LOL(train)→LSRW(eval)|LOL(train)→LOL(eval)|
> |------|------|:--------------------------:|:----------------------:|
> |Supervised|LLFlow|18.68/0.518|24.99/0.871|
> |Supervised|RetinexFormer|19.15/0.529|27.18/0.850|
> |Unsupervised|ZeroDCE|18.87/0.467|20.99/0.596|
> |Unsupervised|ZeroIG [7*]|13.30/0.341|22.17/0.771|
> |Unsupervised|**CLODE**|**20.60/0.557**|23.16/0.752|
> |Unsupervised|**CLODE$\dagger$**|**20.94/0.568**|24.47/0.759|

---

> ### Author Response · Authors · 2024-11-18
> **Response to Reviewer b2uw. (PART 2)**
>
> **W3. More real-world benchmarks**
>
> - Thank you for requesting evaluations on additional benchmark datasets [1*, 2*, 3*, 4*, 6*].
> - As per Reviewer **b2uw**’s request, we evaluated performance on the mentioned benchmark datasets in term of **BRISQUE↓ / Entropy↑**, and included [3*] in the comparison methods. However, due to memory limitations, [3*] could not be evaluated on [5*, 6*]. We have expanded Table 3 in the main manuscript accordingly.
> - **Our method, CLODE demonstrates superior performance on non-reference metrics across all five realistic datasets, except for the BRISQUE value on the DICM dataset (ours is second best on DICM).**
> |Model|DICM [1*]|MEF [2*]|LIME [6*]|NPE [4*]|VV [5*]|
> |-------|:---------:|:--------:|:--------:|:-------:|:------:|
> |Li et al. [3*]|35.92/6.527|34.73/6.544|-|21.47/6.890|-|
> |SCI-easy|**19.83**/6.709|13.42/6.716|16.39/6.748|14.23/7.221|18.93/6.943|
> |SCI-medium|20.97/6.059|14.63/6.934|20.27/7.170|27.42/6.547|21.02/6.601|
> |SCI-difficult|20.26/6.794|13.81/7.098|16.82/7.162|16.49/7.317|18.64/7.212|
> |RUAS|47.01/4.413|34.59/6.248|29.58/6.896|49.83/4.559|51.07/5.191|
> |ZeroDCE	|22.79/6.899|16.22/7.002|17.73/6.967|15.53/7.427|20.98/7.285|
> |ZeroIG [7*]|26.81/6.376|16.47/6.965|19.99/7.155|28.37/7.060|23.07/6.945|
> |**CLODE**|20.30/**7.116**|**12.78/7.383**|**16.26/7.333**|**13.72/7.488**|**18.51/7.296**|
>
> **W4. Scalability**
> - Your question was as follows:
> > “There are many different paradigms for unsupervised methods. Compared with this type of method, is the author's method more scalable? If it can be improved on other methods, I think this article will have higher value.”
> - **Yes. existing curve-adjustment models like ZeroDCE can be easily extended within NODE framework, and CLODE can be applicable to other existing curve adjustment methods.**
> - While it is possible to apply NODE to existing architectures, accurate curve estimation is crucial for high quality. Therefore, we developed a new compact and efficient architecture that can effectively apply on NODE and estimate the fine curves as depicted in Appendix.A.1.2. Figure 7 (a).
> - Regarding the conversion of previous methods to NODE, our experiments with ZeroDCE demonstrated that applying NODE can improve the performance of existing methods. (ZeroDCE_large is a version of the ZeroDCE network with increased parameters for performance enhancement.)
> |Model|#params|PSNR/SSIM (w/o GT Mean)|
> |-------|:--------:|:------------------------------:|
> |ZeroDCE|0.0794|16.49/0.522|
> |ZeroDCE + NODE|0.0794|17.17/0.571|
> |ZeroDCE_large + NODE	|0.3593|19.26/0.637|
> |**CLODE**|**0.2167**|**19.61/0.718**|
> - However, compared to ZeroDCE_large, our proposed architecture, CLODE, achieves superior performance relative to its parameter size. This demonstrates that our network architecture is **more efficient**.
> - **Moreover, our framework can be integrated with the recent RectifiedFlow approach [8\*] to enhance scalability.**
> - RectifiedFlow [8*] transforms the solution paths of Neural ODEs into straight lines, enabling faster estimation of the Neural ODE system. We can first assume that the output and trajectory found by CLODE is the expected optimal solution and then apply CLODE specifically to [8*]. Given the potential to implement advanced Flow Matching methods [8*, 9*], we believe CLODE holds great promise and could achieve rapid inference speeds.
> - **Similarly the integration with vision-language models like CLIP [10\*] will be also feasible.**
> - Unlike other unsupervised methods, CLODE allows users to set a preferred  $t$ , enabling results tailored to user preferences.
> - Using CLIP, it can be employed to find the best-matching image based on a given textual description.
> - For example, let  $h_t$  and  $h_{text}$  denote the features extracted by the CLIP encoder from the  $t$-th image  $I_t$  and the text prompts (e.g., “A well-exposed photo”), respectively. The cosine similarity for the  t -th image  $I_t$  can then be expressed as:
> $$
> sim_{cos}(I_t) = \frac{h_t^T \cdot h_{text}}{||h_t||\cdot||h_{text}||}.$$
> - This cosine similarity metric ($sim_{cos}(\cdot)$), combined with an appropriate threshold, CLODE can be used to automatically generate more user-friendly images without user-intervention.

---

> > ### Author Response · Authors · 2024-11-18
> > **Response to Reviewer b2uw. (PART 3) References**
> >
> > > **References**
> > >
> >
> > [1*] Chulwoo Lee, Chul Lee, and Chang-Su Kim. Contrast enhancement based on layered difference representation of 2d histograms. TIP, 2013.
> >
> > [2*] Kede Ma, Kai Zeng, and Zhou Wang. Perceptual quality assessment for multi-exposure image fusion. TIP, 2015.
> >
> > [3*] Li, Mading, et al. "Structure-revealing low-light image enhancement via robust retinex model." *IEEE Transactions on Image Processing* 27.6, 2018.
> >
> > [4*] Shuhang Wang, Jin Zheng, Hai-Miao Hu, and Bo Li. Naturalness preserved enhancement algorithm for non-uniform illumination images. TIP, 2013
> >
> > [5*] Vassilios Vonikakis, Rigas Kouskouridas, and Antonios Gasteratos. On the evaluation of illumination compensation algorithms. Multimedia Tools and Applications, 2018.
> >
> > [6*] Xiaojie Guo, Yu Li, and Haibin Ling. Lime: Low-light image enhancement via illumination map estimation. TIP, 2016.
> >
> > [7*] Shi, Yiqi, et al. "ZERO-IG: Zero-Shot Illumination-Guided Joint Denoising and Adaptive Enhancement for Low-Light Images." In CVPR, 2024.
> >
> > [8*]  Liu, Xingchao, and Chengyue Gong. "Flow Straight and Fast: Learning to Generate and Transfer Data with Rectified Flow." In ICLR, 2023
> >
> > [9*] Tong, A., Malkin, N., Huguet, G., Zhang, Y., Rector-Brooks, J., Fatras, K., ... & Bengio, Y. "Improving and generalizing flow-based generative models with minibatch optimal transport." In Transactions on Machine Learning Research, 2024
> >
> > [10*] Alec Radford, Jong Wook Kim, Chris Hallacy, Aditya Ramesh, Gabriel Goh, Sandhini Agarwal,
> > Girish Sastry, Amanda Askell, Pamela Mishkin, Jack Clark, et al. Learning transferable visual
> > models from natural language supervision. In *International conference on machine learning*. PMLR, 2021.

---

> > > ### Author Response · Authors · 2024-11-23
> > >
> > > To Reviewer **b2uw**,
> > >
> > > We sincerely thank you for your thoughtful reviews and valuable feedback. We hope that our responses have adequately addressed your concerns. Should you have any further questions after reviewing our rebuttal, please do not hesitate to reach out to us.
> > >
> > > We remain fully committed to addressing any remaining issues during the discussion period.
> > > Thank you once again for your time and consideration.
> > >
> > > Best regards,
> > >
> > > Authors of #4698

---

> > ### Comment · Reviewer_b2uw · 2024-11-26
> >
> > Thanks to the author for the detailed rebuttal, which solved many of my concerns. However, I still have a few questions: For the first table you provided, compared with PairLIE, the improvement seems to be negligible compared to the computational overhead, especially the speed seems to be 7 times worse? In addition, the author said that the more parameters, the easier it is for the network to overfit. The relationship between the two does not seem to be so simple. And some data sets of low-light enhancement tasks have been proven to not be a good proof of the performance of the model, such as LOL v1. What's more, for the comparison of no-reference indicators, Entropy still has some problems. It would be better if it could be compared with some more effective ones, such as NIMA, NIQE, etc.
> >
> > For the value of the entire article, I think it has made a certain contribution. As the author said, its scalability, I think it is a great value. In general, I think this article has defects in experiments and comparisons, but the overall value is worthy of recognition. Therefore, I will temporarily keep my 5 points (close to 5.5 points)

---

> ### Author Response · Authors · 2024-11-26
> **Additional Response to Reviewer b2uw. (PART1)**
>
> Thank you for reviewing the rebuttal letter we prepared and for providing valuable feedback. We have prepared responses to address the additional concerns raised by the reviewer.
>
> We have summarized your additional questions into 4 main points.
>
> - **New-Q1. Balance between performance improvement and computational cost (comparison with PairLIE).**
>
> - **New-Q2. Relationship between the number of parameters and overfitting.**
>
> - **New-Q3. Validity of low-light enhancement dataset like LOLv1.**
>
> - **New-Q4. Results on non-reference metrics such as NIQE and NIMA**
>
>
> **New-Q1. Balance between performance improvement and computational cost**
>
> - The reviewer pointed out that compared to PairLIE, the performance improvement seems negligible, particularly given the computational overhead, with the speed being 7 times slower.
> - We appreciate the reviewer’s insightful comment. As the reviewer pointed out, our proposed method does involve a computational overhead compared to PairLIE. **However, we would like to emphasize that comparing our method might be unfair.**
> - **PairLIE uses multiple levels of low-light brightness images of the same scene as training inputs. By leveraging the strong constraint of training with paired inputs from the same scene**, the model can achieve low-light image enhancement.
> - In contrast, our method follows a commonly used approach in unsupervised low-light image enhancement, utilizing single images as training inputs. Our method imposes no constraints on the training dataset, making it advantageous for training on new datasets.
> - For example, when our model is trained on a new low-light dataset such as RELLISUR [1*], it can achieve better results than those presented in the main manuscript. **In contrast, PairLIE cannot be trained on such new datasets without the user manually creating paired data.**
> |PSNR/SSIM|RELLISUR(train)→LOLv1(eval)|LOLv1(train)→LOLv1(eval)|
> |-------|:---------------------------------:|:----------------------------:|
> |**CLODE**|**19.84/0.779**|	19.61/0.718|
> |PairLIE|***Unable to train***|19.51/0.736|
>
> - Therefore, while we acknowledge that comparing PairLIE with unsupervised methods that use single datasets may not be entirely fair, we included computational aspect in Sec.5 Limitation of the main manuscript to transparently discuss the constraints of our approach.
> - **Overall,  CLODE demonstrates superior performance compared to PairLIE, even with fewer parameters and single-image training. We kindly ask you to reconsider this point.**
> |Model|	Paired train set|	LSRW|	LOLv1|	SICE|	#Params|	Time (S)|
> |-------|:------------------:|:----------:|:----------:|:-------:|:-------------:|:---------:|
> |PairLIE	|o|	16.97/0.498|	19.51/**0.736**|	13.39/0.619|	0.3417|	**0.008**|
> |**CLODE**|	x	|**17.28/0.533**	|**19.61**/0.718	|**15.01/0.687**|	**0.2167**|	0.056|

---

> ### Author Response · Authors · 2024-11-26
> **Additional Response to Reviewer b2uw. (PART2)**
>
> **New-Q2. Relationship between the number of parameters and overfitting.**
>
> - First, while we mentioned the general perspective that fewer parameters may lead to reduced overfitting concerns, we agree with reviewer's point that this issue is not that simple.
> - Therefore, to address the reviewer’s concern, we approach the issue of overfitting from two aspects: **training dynamics** and **performance on other datasets**.
> - **Training dynamics**: To analyze stability during training and the potential for overfitting, we have added *Appendix Sec. A.7. Loss and Metric Curves*, following **the reviewer NMzG’s Question 4.** As shown in *Appendix Figure 19*, the individual losses used in our method decrease steadily during training, indicating stable learning. Additionally, Figure 20 and 21 demonstrates a corresponding steady increase in PSNR and SSIM as the loss decreases. In summary, the gradual reduction of the losses reflects training stability, and the progressive improvement in PSNR and SSIM suggests that overfitting to the training set did not occur.
> - Unlike other models that use the parameter weights corresponding to the highest evaluation metrics during training, we train for 100 epochs and use the results from the 100th epoch as the final parameter weights.
> - While it is possible to interpret the results as a training of the unique characteristics trained specifically from the LOLv1 training, we also provide results on other datasets to support this interpretation.
> - **Results on other datasets**: In ***Appendix Table 12** of our main manuscript*, we present **LPIPS↓** results for CLODE across multiple datasets (LSRW, LOL, SICE, MSEC). These results were obtained by training both CLODE and the comparison methods on LOLv1 and SICE datasets. CLODE exhibits quantitatively superior **image perceptual performance** on average. This highlights that our continuous Neural ODE-based method demonstrates superior generalization compared to other methods.
> - ***Appendix Table 12*, which was initially provided to demonstrate generalization performance on other datasets, is presented again below for your convenience.**
> |Dataset|	URetinexNet|	RetinexFormer|	SCI-medium|	RUAS|	ZeroDCE|	NightEnhancement|	PairLIE|	CLODE|
> |---------|:--------------:|:----------------:|:-------------------:|:------:|:-------------:|:---------------------:|:------------:|:---------:|
> |LSRW|	**0.308**|	0.315|	0.398|	0.469|	0.317|	0.583|	0.342|	0.331|
> |LOL|	**0.121**	|0.131|	0.358|	0.270|	0.335|	0.241|	0.248|	0.263|
> |SICE|	0.264|	0.263|	0.486|	0.608|	0.239|	0.360|	0.305|	**0.235**|
> |MSEC|	0.393|	0.362|	0.396|	0.668|	0.329|	0.462|	0.431|	**0.223**|
> |Average|	0.272|	0.268|	0.410|	0.504|	0.305|	0.412|	0.332|	**0.263**|
>
> **New-Q3. Validity of low-light enhancement dataset like LOLv1.**
>
> - We fully acknowledge, as the reviewer pointed out, that strong performance on LOLv1 alone does not conclusively demonstrate a robust low-light image enhancement model.
> - However, since previous unsupervised methods primarily focused on LOLv1, we also utilized LOLv1 in our study.
> - **Additionally, as mentioned in New-Q1 and New-Q2, CLODE demonstrates superior performance compared to other methods on LSRW, SICE, and MSEC datasets.**
>
> **New-Q4. Results on non-reference metrics such as NIQE and NIMA**
>
> - Thank you for suggesting appropriate non-reference metrics. We present the results for **NIQE$\downarrow$** and **NIMA$\uparrow$** as shown below.
> - Excluding the NIMA score on the LIME dataset, CLODE demonstrates the best performance.
> |Model|DICM|MEF|LIME|NPE|VV|
> |-------|----------|---------|------------|----------|---------|
> |SCI-easy|	3.902/4.557|	3.655/5.150|	4.118/4.737|	4.022/4.622|	2.903/3.990|
> |SCI-medium|	4.129/4.430|	3.619/5.016|	4.211/4.566|	4.321/4.367|	2.818/3.900|
> |SCI-difficult|	4.031/4.415|	3.664/4.959|	4.097/4.446|	4.166/4.141|	2.903/3.990|
> |RUAS|	7.153/4.178|	5.408/4.732|	5.420/4.391|	7.063/4.171|	5.230/4.015|
> |ZeroDCE|	3.741/4.563|	3.310/5.190|	3.786/**4.597**|	3.946/4.698|	2.585/3.849|
> |ZeroIG|	3.980/4.559|	3.764/4.988|	4.441/4.529|	4.662/4.351|	2.779/4.124|
> |PairLIE|	4.297/4.360|	4.203/4.702|	4.547/4.389|	4.238/4.509|	3.295/4.100|
> |CLODE|	**3.628/4.778**|	**3.286/5.287**|	**3.650**/4.510|	**3.888/4.738**|	**2.770/4.133**|

---

> > ### Comment · Reviewer_b2uw · 2024-11-26
> >
> > Thanks for your reply. However, why can the normal decline of the loss curves of training dynamics prove that there is no overfitting? In this case, isn't it more convincing to observe the loss of testing dynamics? Isn't the normal decrease in training loss the normal state of training? This proof seems to be problematic. Are the curves of other methods the same?

---

> > > ### Author Response · Authors · 2024-11-26
> > >
> > > Thank you for your prompt response, and we apologize for any confusion caused!
> > >
> > > - To examine the test loss dynamics, we re-trained the model, which took some time. We apologize for the delay in providing our response.
> > >
> > > - **As you suggested, presenting test dynamics would be more appropriate.** **We have now added the loss curves of test dynamics to Figure 22 in the updated rebuttal version for greater clarity** and kindly ask you to refer to it.
> > >
> > > - **The loss curves of testing does not show any tendency toward overfitting.** While the reductions in $\mathcal L_{col}$, $\mathcal L_{exp}$, and $\mathcal L_{spa}$ are minimal, they serve as constraints. Details regarding the ablation study on the loss functions can be found in **Appendix Sec. A2, Impact of Each Loss Function**, with the results presented in **Figure 8** and **Table 7**.
> > >
> > > - Additionally, under the same dataset training conditions, and considering the robust performance across multiple datasets, we would like to assert that our method demonstrates better generalization capabilities compared to other approaches.

---

> > > > ### Comment · Reviewer_b2uw · 2024-11-28
> > > >
> > > > Thank you for your reply. In my opinion, most of my questions have been addressed, but I still think that the final version needs further improvement and discussion along the lines of the rebuttal section. Overall balancing the contributions and shortcomings, I am willing to revise my rating to 6 score.

---

> > > > > ### Author Response · Authors · 2024-11-28
> > > > > **To reviewer b2uw**
> > > > >
> > > > > **To reviewer b2uw**,
> > > > >
> > > > > Thank you for your insightful and constructive feedback throughout the review process.
> > > > >
> > > > > Since the rebuttal PDF update period has passed, we will incorporate additional details addressing the concerns you raised, based on your insightful review, into the final version of the paper. Thank you once again for your thoughtful feedback!
> > > > >
> > > > > Best regards,
> > > > >
> > > > > Authors of #4698

---

> ### Author Response · Authors · 2024-11-26
> **Additional Response to Reviewer b2uw. (PART3)**
>
> Lastly, thank you for positively evaluating our scalability. Thanks to the reviewer’s comments, we were able to further enhance the experiments in the manuscript. Additionally, through the rebuttal, we have included ZeroIG, introduced at CVPR 2024, as a comparison method.
>
> We also agree that LOLv1 is not an ideal dataset for low-light evaluation. However, since most comparison methods primarily use LOLv1, we have followed this approach. In addition, to address this limitation, as shown in our responses to additional questions, we have included evaluations on multiple datasets in the appendix of our paper initially. We humbly hope that this point will also be taken into consideration.
>
> Please review our responses once again, and if you have any additional concerns, do not hesitate to reach out with further questions. We look forward to your feedback!
>
> > **Reference**
> >
> [1*] Aakerberg, Andreas, Kamal Nasrollahi, and Thomas B. Moeslund. "RELLISUR: A real low-light image super-resolution dataset." in NeurIPS 2021.

---

### Author Response · Authors · 2024-11-18

**To the reviewers,**

Thank you for taking the time to review our paper and provide valuable feedback. Your insightful comments and suggestions have been instrumental in improving our proposed method.

In response to your feedback, we have updated our paper and reuploaded the revised version. The key changes are as follows:

1. **Figure 2 (Flowchart):** We have enhanced the overall flowchart by incorporating detailed network information previously provided in the Appendix. **(NMzG W3)**
2. **Sec. 6 (Conclusion):** We refined the conclusion to articulate our contributions more clearly and precisely. **(NMzG W4)**
3. **Sec. A.7 (Training Dynamics):** We added a new section on training dynamics, including details on loss function behavior (**Figure 19**) and performance metrics (**Figure 20 and 21**). This addition aims to provide a better understanding of the training process. **(NMzG Q4)**

We hope these updates address your concerns and further enhance the quality of our paper. Thank you once again for your constructive feedback and for considering our revisions.

---

### Author Response · Authors · 2024-11-25

Dear Reviewers,

Thank you for your efforts and valuable feedback so far. We are especially grateful to **NMzG** for your active engagement during this process.

We kindly request reviewers **b2uw** and **Asgg** to share your thoughts on our responses as well.
As the discussion period is approaching its end, with a deadline set for **Nov. 26th**, we are still awaiting responses from reviewers who have not yet provided feedback. We have carefully prepared specific responses to address each concern raised.

We would greatly appreciate it if you could review our responses and let us know if there is anything unclear or that could be further detailed to enhance the quality of our paper. Your input would be invaluable in helping us improve.

Best regards,

Authors of #4698

---

### Meta-Review · Area_Chair_BtUR · 2024-12-19

**Metareview:**

This paper utilize the neural ODEs and develops a continuous exposure learning approach for low-light image enhancement. Experimental results show the effect of the proposed method.

The major concerns of reviewers include the computational cost, generalization ability, and limited evaluations. In the rebuttal, the author solve the most concerns of reviewers. However, there are still major issues as detailed below.

The paper claim that the convergence of the existing method are not guaranteed and the proposed method has better convergence  property. However, this is not explained or demonstrated in the paper. In addition, no theoretical analysis is provided. That is, the contribution of the paper has been exaggerated.

Based on the comments of reviewers and the above-mentioned issues, the recommendation of this paper is borderline.

**Additional Comments On Reviewer Discussion:**

The major concerns of reviewers include the computational cost, generalization ability, and limited evaluations. In addition, as mentioned above, the authors overclaim the contributions of this paper.

---

### Decision · Program_Chairs · 2025-01-22

Accept (Spotlight)